

# Computation and analysis of atmospheric carbon dioxide annual mean growth rates from satellite observations during 2003-2016

Michael Buchwitz[1], Maximilian Reuter[1], Oliver Schneising[1], Stefan Noël[1], Bettina Gier[1,2], Heinrich Bovensmann[1], John P. Burrows[1], Hartmut Boesch[3,4], Jasdeep Anand[3,4], Robert J. Parker[3,4], Peter Somkuti[3,4], Rob G. Detmers[5], Otto P. Hasekamp[5], Ilse Aben[5], André Butz[2,6], Akihiko Kuze[7], Hiroshi Suto[7], Yukio Yoshida[8], David Crisp[9], Christopher O'Dell[10]

[1]Institute of Environmental Physics (IUP), University of Bremen, Bremen, Germany

[2]Institute of Atmospheric Physics, Deutsches Zentrum für Luft- und Raumfahrt (DLR), Oberpfaffenhofen, Germany

[3]Earth Observation Science, University of Leicester, Leicester, UK

[4]NERC National Centre for Earth Observation, Leicester, UK

[5]SRON Netherlands Institute for Space Research, Utrecht, The Netherlands

[6]Meteorologisches Institut, Ludwig-Maximilians-Universität (LMU), Munich, Germany

[7]Japan Aerospace Exploration Agency (JAXA), Tsukuba, Japan

[8]National Institute for Environmental Studies (NIES), Tsukuba, Japan

[9]Jet Propulsion Laboratory (JPL), Pasadena, CA, USA

[10]Colorado State University (CSU), Fort Collins, CO, USA

*Correspondence to*: Michael Buchwitz (Michael.Buchwitz@iup.physik.uni-bremen.de)

**Abstract.** The growth rate of atmospheric carbon dioxide ($CO_2$) reflects the net effect of emissions and uptake resulting from anthropogenic and natural carbon sources and sinks. Annual mean $CO_2$ growth rates have been determined globally and for selected latitude bands from satellite retrievals of column-average dry-air mole fractions of $CO_2$, i.e., $XCO_2$, for the years 2003 to 2016. The global $XCO_2$ growth rates agree with National Oceanic and Atmospheric Administration (NOAA) growth rates from $CO_2$ surface observations within the uncertainty of the satellite-derived growth rates (mean difference ± standard deviation: 0.0±0.24 ppm/year; R: 0.87). This new and independent data set confirms record large growth rates around 3 ppm/year in 2015 and 2016, which are attributed to the 2015/2016 El Niño. Based on a comparison of the satellite-derived growth rates with human $CO_2$ emissions from fossil fuel combustion and with El Niño Southern Oscillation (ENSO) indices, we estimate by how much the impact of ENSO dominates the impact of fossil fuel burning related emissions in explaining the variance of the atmospheric $CO_2$ growth rate.




## 1 Introduction

Atmospheric carbon dioxide ($CO_2$) is an important greenhouse gas that causes global warming (IPCC 2013). Sources that emit $CO_2$ into the atmosphere include anthropogenic and natural sources at the surface, and the

oxidation of carbon monoxide and hydrocarbons in the atmosphere. The sinks that remove $CO_2$ primarily at the surface include biological (photosynthesis), physical (solubility) and some geologic processes (e.g., carbonate weathering). Anthropogenic emissions of $CO_2$, primarily from fossil fuel combustion, have increased the atmospheric $CO_2$ mixing ratios at the surface by more than 40% since pre-industrial times, from less than 280 parts per million (ppm) to 402.8±0.1 ppm in 2016 (Dlugokencky and Tans, 2017a). A global increase of

atmospheric $CO_2$ by 1 ppm in a one-year time period corresponds to an annual increase of 2.12 GtC/year (Ballantyne et al., 2012; Prather et al., 2012). However, this increase in mass does not directly correspond to the emissions. The reason is that only a fraction of the emitted $CO_2$ remains in the atmosphere as $CO_2$ is partitioned between the atmosphere and ocean and land carbon sinks. On average, somewhat less than half of the emitted $CO_2$ remains in the atmosphere but this "airborne fraction" varies substantially from year to year (Le Quéré et al.,

2016, 2017). Variations of the airborne fraction are not well understood primarily because of an inadequate understanding of the terrestrial carbon sink, which introduces large uncertainties for climate prediction (e.g., IPCC 2013; Peylin et al., 2013; Wieder et al., 2015; Huntzinger et al., 2017). Identification of the origin of changes of the growth rate requires additional information for the attribution to particular sources or sinks (Peters et al., 2017). Atmospheric $CO_2$ growth rates inferred from in-situ $CO_2$ surface measurements are regularly

determined and published, for example, by the National Oceanic and Atmospheric Administration (NOAA) (see https://www.esrl.noaa.gov/gmd/ccgg/trends/gr.html). In this study, we present and interpret atmospheric growth rates determined from the remote sensing of $CO_2$ vertical columns from space, which are described in the following section.

## 2 Global satellite observations of atmospheric $CO_2$ columns

Satellites provide retrievals of $CO_2$ vertical columns in terms of the $CO_2$ column-average dry-air mole fraction, denoted $XCO_2$. Although a relatively new field, satellite-based $XCO_2$ data products have already been used to improve our knowledge of natural (e.g., Basu et al., 2013; Maksyutov et al., 2013; Chevallier et al., 2014; Reuter et al., 2014a; Schneising et al., 2014; Houweling et al., 2014; Parker et al., 2016; Heymann et al., 2017; Liu et al.,

2017; Kaminski et al., 2017) and anthropogenic (e.g., Schneising et al., 2013; Reuter et al., 2014b; Kort et al., 2012; Hakkarainen et al., 2016; Nassar et al., 2017) $CO_2$ sources and sinks but only a few studies explicitly present and discuss $CO_2$ growth rates. Buchwitz et al., 2007, analyzed the first three years (2003-2005) of $XCO_2$ retrievals from SCIAMACHY/ENVISAT (Burrows et al., 1995; Bovensmann et al., 1999) generated using the WFM-DOAS retrieval algorithm (Buchwitz et al., 2006). They computed year-to-year $CO_2$ variations and

compared the $XCO_2$ increase with the $XCO_2$ increase computed from the output of NOAA's $CO_2$ assimilating system CarbonTracker (Peters et al., 2007) and found agreement within 1 ppm/year. Schneising et al., 2014, computed growth rates from the 2003-2011 SCIAMACHY $XCO_2$ record. They compared the derived annual growth rates with surface temperature and found that years having higher temperatures during the vegetation



growing season are associated with larger growth rates in atmospheric $CO_2$ at northern mid-latitudes. Growth rates from GOSAT (Kuze et al., 2016) are published by the National Institute for Environmental Studies (NIES), Tsukuba, Japan (NIES 2017).

In this study, we analyze a new satellite $XCO_2$ data set covering 14 years (2003-2016) generated from
SCIAMACHY/ENVISAT and TANSO-FTS/GOSAT. We use the $XCO_2$ data product Obs4MIPs (Observations for Model Intercomparisons Project) version 3 (O4Mv3), which is a gridded (Level 3) monthly data product at 5º latitude by 5º longitude spatial resolution in Obs4MIPs format (Buchwitz et al., 2017a). Obs4MIPs (https://www.earthsystemcog.org/projects/obs4mips/) is an activity to make observational products more accessible for climate model intercomparisons (e.g., Lauer et al., 2017). The O4Mv3 $XCO_2$ data product was
generated by gridding (averaging) the $XCO_2$ Level 2 (i.e., individual soundings) product generated with the Ensemble Median Algorithm (EMMA, Reuter et al., 2013). EMMA uses as input an ensemble of $XCO_2$ Level 2 data products (Buchwitz et al., 2015, 2017a, 2017b; Reuter et al., 2013) from SCIAMACHY/ENVISAT and TANSO-FTS/GOSAT. To generate the O4Mv3 product, the EMMA version 3.0 (EMMAv3, Reuter et al., 2017e) product was used. The list of satellite products used for the generation of the EMMAv3 Level 2 product - and
therefore also for the O4Mv3 Level 3 data product used in this study - is provided in Tab. 1. The quality of this product relative to Total Carbon Column Observing Network (TCCON) ground-based observations (Wunch et al., 2011, 2015) can be summarized as follows (Buchwitz et al., 2017c): +0.23 ppm overall (global) bias, relative accuracy 0.3 ppm (1-sigma), and very good stability in terms of linear bias trend (-0.02±0.04 ppm/year).

Figure 1 presents an overview of the O4Mv3 product in terms of time series and global $XCO_2$ maps. The maps
show the typical coverage of $XCO_2$ from SCIAMACHY (until April 2012) and GOSAT (since mid 2009). As can be seen, the time series for the three latitude bands shown in Fig. 1 have very similar slopes. They mainly differ in the amplitude of the seasonal cycle, which reflects the latitudinal dependence of uptake and release of atmospheric $CO_2$ by the terrestrial biosphere (Schneising et al., 2014). These time series have been used to compute annual mean $CO_2$ growth rates as will be explained in the following section.


## 3 Atmospheric $CO_2$ growth rates from satellite observations

National Oceanic and Atmospheric Administration (NOAA) defines the annual mean $CO_2$ growth rate for a given year as the $CO_2$ concentration difference at the end of that year minus the $CO_2$ concentration at the beginning of that year (Thoning et al., 1989; see also additional explanations as given on the NOAA/ESRL website
(https://www.esrl.noaa.gov/gmd/ccgg/about/global_means.html)). We adopt this definition. As described below, our method involves the following three steps: (i) Computation of an $XCO_2$ time series (at monthly resolution and sampling) by averaging the $XCO_2$ in the region of interest. (ii) Computation of monthly sampled $XCO_2$ annual growth rates by computing the difference of the $XCO_2$ value of month i minus the $XCO_2$ value of month i-12 and computation of the corresponding uncertainty estimate. (iii) Computation of annual mean growth rates and
their corresponding uncertainties from the monthly sampled annual growth rates.

In the following, this method is described in detail using Fig. 2 for illustration. Figure 2 shows how the growth rates are computed for the latitude band 30ºN-60ºN, i.e., for northern mid-latitudes. In Figure 2a monthly satellite





$XCO_2$ (O4Mv3), as obtained by averaging all the individual (5°x5°) $XCO_2$ values in the selected latitude band, is plotted. As can be seen, the computed time series does not start at the beginning of 2003 but in April 2003. As explained in Buchwitz et al., 2017d (see discussion of their Fig. 6.1.1.1) the underlying SCIAMACHY BESD v02.01.02 $XCO_2$ data product (see Tab. 1) apparently suffers from an approximately 1 ppm high bias in the first

few months of 2003. The exact magnitude of this bias has not been quantified due to lack of TCCON validation data in this early time period. As this bias in early 2003 is critical for the year 2003 growth rate, we have omitted the first three months of 2003 for the computation of the growth rates shown in this publication.

Figure 2b shows monthly sampled annual growth rates as computed from the monthly $XCO_2$ values shown in Fig. 2a. Each value is the difference of two monthly $XCO_2$ values corresponding to the same month (e.g.,

January) but different years (e.g., 2004 and 2005). For example, the first data point (first diamond symbol) shown in Fig. 2b is the difference of the April 2004 $XCO_2$ value minus the April 2003 $XCO_2$ value. The second data point corresponds to May 2004 minus May 2003, etc. The time difference between the monthly $XCO_2$ pairs is always one year and the time assigned to each $XCO_2$ difference is the time in the middle of that year. Therefore, the time series shown in Fig. 2b starts six months later and ends six months earlier as compared to the time series

shown in Fig. 2a. Each $XCO_2$ difference shown in Fig. 2b therefore corresponds to an estimate of the $XCO_2$ annual growth rate and the position on the time-axis corresponds to the middle of the corresponding one-year time period.

A 1-sigma uncertainty estimate has been computed for each of the monthly sampled annual growth rates shown in Fig. 2b (see grey vertical bars). They have been computed such that they reflect the following aspects: (i) the

standard error of the O4Mv3 $XCO_2$ values as given in the O4Mv3 data product file for each of the 5°x5° grid cells, (ii) the spatial variability of the $XCO_2$ within the selected region, (iii) the temporal variability of the annual growth rates in the one year time interval, which corresponds to the annual growth rate, and (iv) the number of months (N) with data located in that one year time interval. The uncertainties have been computed as the mean value of three terms divided by the square root of N. The first term is the mean value of the standard error, the

second term is the standard deviation of the $XCO_2$ values in the selected region and the third term is the standard deviation of the monthly sampled annual growth rates in the corresponding one-year time interval.

Figure 2c shows the final result, i.e., the annual mean $XCO_2$ growth rates and their estimated (1-sigma) uncertainties. The annual mean growth rates have been computed by averaging all the monthly sampled annual growth rates (shown in Fig. 2b), which are located in the year of interest (e.g., 2003). For most years, 12 annual

growth rate values are available for averaging but there are some exceptions. For example, for the year 2003 only 3 values are present as can be seen from Fig. 2b and for the years 2014 and 2015 there are only 11 values as no data are available for January 2015 due to issues with the GOSAT satellite. The uncertainty of the annual mean growth rate has been computed by averaging the uncertainties assigned to each of the monthly sampled annual growth rates (shown as grey vertical bars in Fig. 2b) scaled with a factor, which depends on the number of

months (N) available for averaging. This factor is the square root of 12/N. It ensures that the uncertainty is larger, the less data points are available for averaging. Overall, our uncertainty estimate is quite conservative, as we do not assume that errors improve upon averaging. As a result of this procedure, the error bar of the year 2003 growth rate is quite large (0.69 ppm/year, see Tab. 2, where all numerical values are listed). This is because the



monthly sampled annual growth rate varies significantly in 2003 (see Fig. 2b) and because only N=3 data points are available for averaging in 2003. In contrast, the year 2005 growth rate uncertainty is much smaller (0.25 ppm/year) because the growth rates vary only little during 2005 and because N=12 data points are available for averaging.

Figure 3 shows the corresponding results for the global data set. As can be seen, all time series are similar to the ones shown in Fig. 2 for northern mid-latitudes. However, there are also difference, e.g., the seasonal cycles as shown in Fig. 2a and Fig. 3a. For northern mid-latitudes (Fig. 2a) the shape of these cycles is very similar for all years in contrast to the global data shown in Fig. 3a. This is due to spatial sampling differences as the first few years (until 2008) are "land only" data as the SCIAMACHY $XCO_2$ is limited to observations over land whereas

GOSAT $XCO_2$ (from 2009 onwards) is not restricted to land (see global maps shown in Fig. 1). For the northern mid-latitude region the land coverage dominates (see global map in Fig. 2a). Therefore, for northern mid-latitudes SCIAMACHY and GOSAT sample similar regions, in contrast to the global region (Fig. 3), where the spatial sampling differences are larger. In Fig. 3c also the NOAA global growth rates (Dlugokencky and Tans, 2017b) are shown. As can be seen, the satellite-derived growth rates agree well with the NOAA growth rates obtained

from $CO_2$ surface observations. For the time period 2003-2016 the linear correlation coefficient R is 0.87 and the difference is 0.0±0.24 ppm/year (mean difference ± standard deviation).

Growth rate time series for several latitude bands are shown in Fig. 4. As can be seen from Fig. 4, the growth rates are similar in all latitude bands including the global results (for numerical values see Tab. 2). The reason for this is that atmospheric $CO_2$ is long-lived and therefore well-mixed. As can also be seen from Fig. 4, the largest

growth rates are approximately 3 ppm/year during 2015 and 2016. These record large growth rates (Peters et al., 2017) are attributed to the consequences of the strong 2015/2016 El Niño event, which produced large $CO_2$ emissions from fires and enhanced net biospheric respiration in the tropics relative to normal conditions (Heymann et al., 2017; Liu et al., 2017). Many of these fires are initiated by humans, for example, to clear tropical forests. In this study, human emissions of $CO_2$ are defined as emissions from fossil fuel combustion and

industry (Le Quéré et al., 2016, 2017) but do not include, for example, $CO_2$ emissions originating from slash and burn agriculture.

## 4   Correlation of $CO_2$ growth rates with fossil $CO_2$ emissions and ENSO indices

Figure 5 shows a comparison of the $CO_2$ annual mean growth rates (Fig. 5a) with annual global $CO_2$ emissions

from fossil fuel combustion and industry (Fig. 5b) (Le Quéré et al., 2017; GCP 2017) (correlation of growth rate and human emissions: $R^2 = 21\%$). As can be seen, the growth rates vary significantly in recent years despite nearly constant human emissions. Figure 5d shows two ENSO indices: the Southern Oscillation Index (SOI, blue lines) (NOAA 2017a; Ropelewski and Jones, 1987) and the Oceanic Niño Index (ONI, green lines) (NOAA 2017b). Whereas SOI is defined as the normalized pressure difference between Tahiti and Darwin (values less

than -1 indicate the presence of a strong El Niño), ONI is based on Sea Surface Temperature (SST) differences (positive values correspond to El Niño). The dotted lines correspond to the original (i.e., unshifted) annual mean indices and the solid lines correspond to time shifted ENSO indices. Time shifts have been investigated to consider the delay in atmospheric response to ENSO-induced changes. As shown in Fig. 5c, the growth rate



response as quantified by $R^2$ is largest after 4 months for ONI and after 7 months for SOI ($R^2$ = 37% for both cases). These maxima have been adopted for the shifted lines in Fig 5d. This finding is consistent with results from other studies, where lags in the range 3-9 months have been reported (Jones et al., 2001; Chylek et al., 2018).

In order to quantify the impact of the human $CO_2$ emissions and of ENSO, as described by the two indices SOI and ONI, on growth rate variations, we employ the method of "variation partitioning" (Peres-Neto et al., 2006). We have fitted three basis functions to the 2003-2016 growth rate time series via linear least-squares minimization (we explain the method in this paragraph using SOI but the method does not depend on which ENSO index is used): (i) a constant offset (variance zero), (ii) the human $CO_2$ emissions (Fig. 5b) and (iii) SOI

shifted by 7 months (blue solid line in Fig. 5d). The variance of the scaled emission, i.e., of the human emission scaled with the corresponding fit parameter, is 0.0743 ppm$^2$/year$^2$ (note that in this section we report numerical values with four digital places but this shall not imply that all decimal places are significant). The variance of the scaled SOI is 0.1128 ppm$^2$/year$^2$ and the variance of the fit residual is 0.0740 ppm$^2$/year$^2$. The sum of the three individual variances is 0.2611 ppm$^2$/year$^2$ whereas the variance of the annual mean growth rate is 0.2357

ppm$^2$/year$^2$. This shows that the sum of the variances is 10.8% larger than the variance of the growth rate, i.e., the sum of the variances is not exactly equal to the variance of the sum. The reason for this is that the $CO_2$ emission and the SOI time series are not uncorrelated (R = -0.14). To account for correlations, we subtract the variance of the residual from the variance of the growth rate. The result is the part of the variance to be explained by the emissions and by the SOI. The ratio of this to be explained variance (0.1617 ppm$^2$/year$^2$) and the sum of the

variances of the emissions and SOI ((0.0743 + 0.1128) ppm$^2$/year$^2$ = 0.1871 ppm$^2$/year$^2$) is 0.8645. The latter is then used as a scaling factor applied to the variances of the emissions and of the SOI. The scaled variances are 0.0642 ppm$^2$/year$^2$ for the emissions and 0.0975 ppm$^2$/year$^2$ for SOI (note that the sum of these scaled variances and the variance of the residual is equal to the variance of the growth rate). From this we conclude that the human emissions explain 27% (= 0.0642/0.2357) of the variance of the growth rate and that ENSO as quantified by the

SOI explains 41% (= 0.0975/0.2357). We computed (1-sigma) uncertainties of these estimates by numerically perturbing the satellite-derived annual mean growth rates taking into account their uncertainty (see Fig. 4) and by subsequently repeating the computations as explained above 10,000 times. The perturbations correspond to random perturbations of the annual mean growth rates assuming normal distributions for each year and no correlation between the different years. This analysis yields that 41±13% of the growth rate variation results from

the impact of ENSO and that 27±14% is due to the human emissions of $CO_2$. Using these simulations, we also computed the fraction of cases where the ENSO impact dominates over the human emissions. This fraction is 66% in this case, i.e., when using SOI and when the analysis is applied to the entire time period 2003-2016. This fraction is interpreted as the probability that ENSO-induced impacts on the variation of the growth rate dominates that of human emissions.

When using ONI instead of SOI, ENSO explains 38±14% of the growth rate variance during 2003-2016, human emissions explain 22±13% and the fraction where ENSO dominates is again 66%. When restricting the time period to 2010-2016, which is dominated by strong 2010/2012 La Niña events (Boening et al., 2012; Rodrigues et al., 2014) and by the strong 2015/2016 El Niño, the results are the following: Using the SOI analysis, we find that ENSO explains 59±19% of the variance, human emissions explain 1±8% and the probability that ENSO




dominates is 95%. For the ONI analysis, we find that ENSO explains 59±20% of the variance, human emissions explain 2±8% and the probability that ENSO dominates is 94%. This analysis shows that the ENSO impact on $CO_2$ growth rate variations dominates over that of human emissions throughout the period 2003-2016 but in particular in the second half of this period, i.e., during 2010-2016.

## 5 Conclusions

We presented a method for the computation of atmospheric $CO_2$ column annual mean growth rates from satellite $XCO_2$ retrievals. The satellite $XCO_2$ data product used is the Obs4MIPs version 3 (O4Mv3) $XCO_2$ data product

based on SCIAMACHY/ENVISAT and TANSO-FTS/GOSAT satellite data. This product covers the time period 2003-2016 and has monthly time and 5°x5° spatial resolution.

The presented method has been applied to the global satellite data and to selected latitude bands. The estimated uncertainty of the satellite-derived annual mean growth rates is typically in the range 0.3-0.5 ppm/year (1-sigma). The global growth rates agree with NOAA within the uncertainty of the satellite-derived growth rates (mean

difference ± standard deviation: 0.0±0.24 ppm/year; R: 0.87). In agreement with NOAA, we find that the growth rates are largest in the years 2015 and 2016. These growth rates are around 3 ppm/year and are attributed to the 2015/2016 El Niño resulting in large $CO_2$ emissions from fires and enhanced net biospheric respiration in the tropics relative to normal conditions (Heymann et al., 2017; Liu et al., 2017). Our analysis also shows that the ENSO impact on $CO_2$ growth rate variations dominates over that of human emissions throughout the period

2003-2016 (14 years) but in particular during the period 2010-2016 (second half of the investigated time period) due to strong La Niña and El Niño events. We estimate the probability that the impact of ENSO on the variability is larger than the impact of human emissions to be 66% for the time period 2003-2016. If the time period is restricted to 2010-2016 this probability increases to 94-95%.

In the future, we plan to regularly update the satellite-derived $XCO_2$ growth rates to monitor this important

quantity. This will also include satellite $XCO_2$ retrievals from other satellite instruments such as $XCO_2$ from NASA's OCO-2 mission (e.g., Eldering et al., 2017; Reuter et al., 2017c, 2017d).



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



**Acknowledgements**

This study has been funded in parts by the European Space Agency (ESA) (via the GHG-CCI project of ESA's
Climate Change Initiative (CCI, http://www.esa-ghg-cci.org/)), by the European Union (EU) (via the Copernicus

Climate Change Service (C3S, https://climate.copernicus.eu/) managed by the European Centre for Medium-
range Weather Forecasts (ECMWF)) and by the State and the University of Bremen. The University of Leicester
GOSAT retrievals used the ALICE High Performance Computing Facility at the University of Leicester. We
thank ESA/DLR for providing us with SCIAMACHY Level 1 data products and JAXA for GOSAT Level 1B
data. We also thank ESA for making these GOSAT products available via the ESA Third Party Mission archive.

We thank NIES for the operational GOSAT $XCO_2$ Level 2 product and the NASA/ACOS team for the GOSAT
ACOS Level 2 $XCO_2$ product. We also thank NOAA for the global $CO_2$ growth rates (file
ftp://aftp.cmdl.noaa.gov/products/trends/co2/co2_gr_gl.txt; access: 24-Nov-2017). The fossil fuel and industry
$CO_2$ emissions have been obtained from the Global Carbon Project website
(http://www.globalcarbonproject.org/carbonbudget/17/data.htm; access: 20-Nov-2017). The Southern Oscillation

Index (SOI) data have been obtained from NOAA (file
https://www.esrl.noaa.gov/psd/gcos_wgsp/Timeseries/Data/soi.long.data; access: 20-Nov-2017). The Oceanic
Niño Index (ONI) data have also been obtained from NOAA (file
https://www.esrl.noaa.gov/psd/data/correlation/oni.data).

**Author contributions**

M.B. supported by O.S., M.R., H.Bov., S.N., B.G., J.P.B.: design, data analysis, interpretation and writing of the
paper. The paper has been significantly improved by: H.Boe., J.A., R.J.P., P.S., R.G.D., O.P.H., I.A., A.B., A.K.,
H.S., Y.Y., D.C., C.O'D. Satellite input data have been provided by: M.R., M.B., O.S. (SCIAMACHY products)
and H.Boe., J.A., R.J.P., P.S., R.G.D., O.P.H., I.A., A.B., A.K., H.S., Y.Y., D. C., C.O'D. (GOSAT products).


**Data availability.** The O4Mv3 $XCO_2$ data product (but also the underlying EMMAv3 product and those
individual sensor Level 2 input products which have been generated with European retrieval algorithms) will be
available (in April/May 2018) via the Copernicus Climate Change Service (C3S, https://climate.copernicus.eu/)
Climate Data Store (CDS). Earlier versions are available from the GHG-CCI website (http://www.esa-ghg-
cci.org/) of the European Space Agency (ESA) Climate Change Initiative (CCI, e.g., Obs4MIPs version 2
(O4Mv2) covering the years 2003-2015).

**Competing financial interests**

The authors declare no competing financial interests.


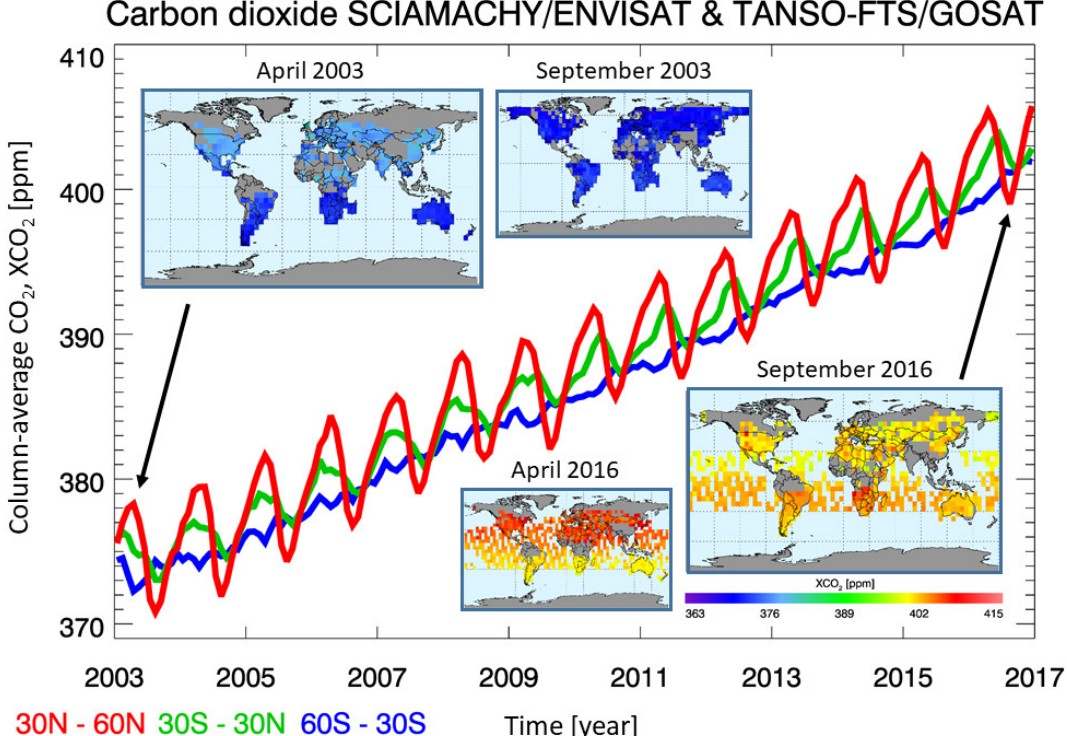

**Figure 1.** Time series and global maps of satellite-derived column-average dry-air mole fractions of carbon
dioxide, i.e., $XCO_2$. Shown is data product Obs4MIPs version 3 (O4Mv3) based on an ensemble of
SCIAMACHY/ENVISAT (until April 2012) and TANSO-FTS/GOSAT (since mid 2009) individual sensor /
individual soundings (Level 2) data products. The three time series correspond to three latitude bands: 30ºN-60ºN
(red), 30ºS-30ºN (green) and 60ºS-30ºS (blue). The maps in the top left show monthly $XCO_2$ for April and
September 2003 (SCIAMACHY, land only) and the maps on the bottom right show monthly $XCO_2$ for April and
September 2016 (TANSO-FTS, land and ocean glint).





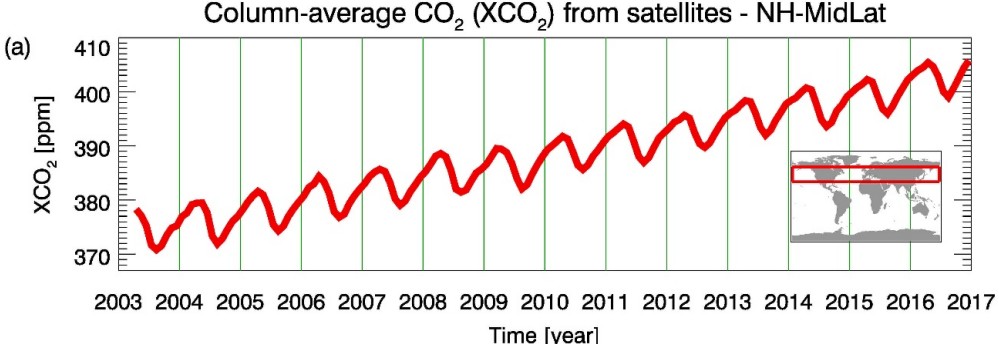

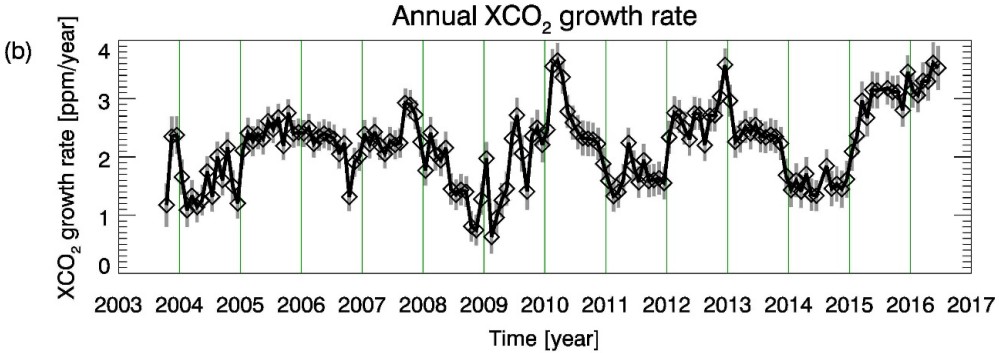

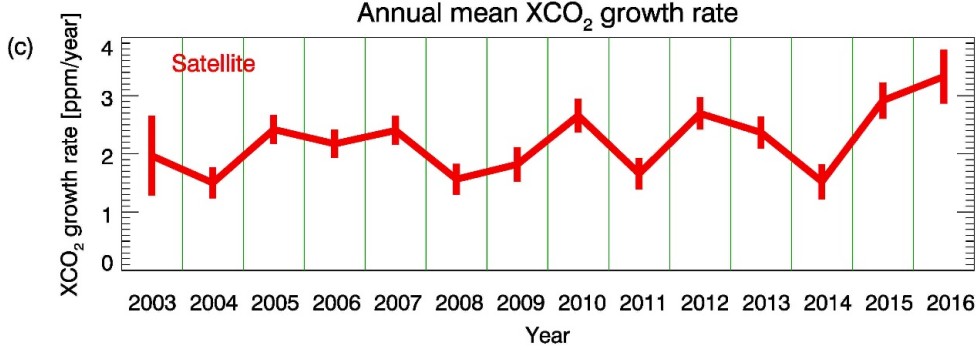

**Figure 2.** Atmospheric $CO_2$ and corresponding growth rates for northern mid-latitudes. (a) Monthly mean $XCO_2$
(red line) for northern mid-latitudes obtained from averaging $XCO_2$ data product O4Mv3 in the latitude band
30ºN-60ºN (see red rectangle in global map). (b) Monthly sampled annual $CO_2$ growth rates as computed from
the red curve shown in (a) including 1-sigma uncertainty (grey vertical bars). (c) Annual mean growth rates
computed from averaging the values shown in (b) including 1-sigma error estimates (vertical bars) (the numerical
values are listed in Tab. 2).





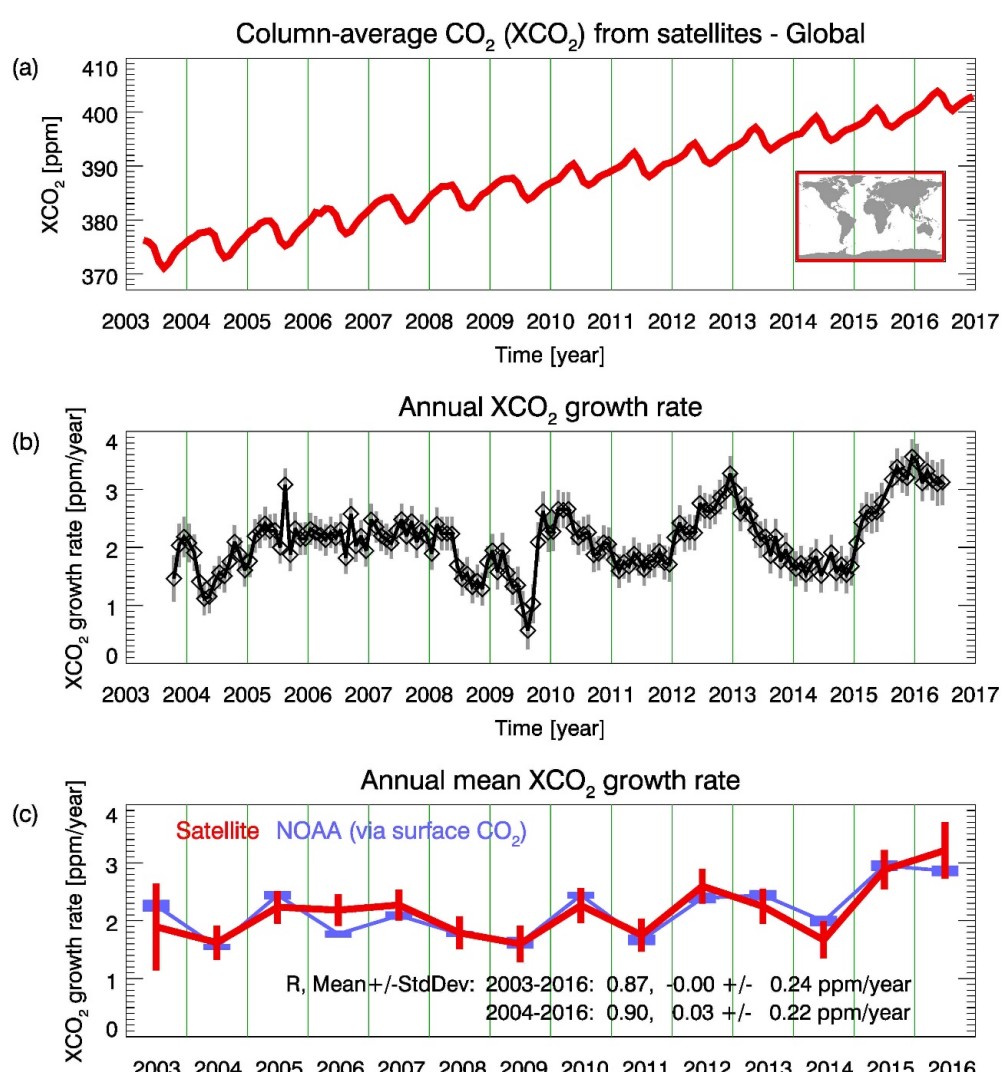

**Figure 3. As Fig. 2 but for the entire globe.** The NOAA annual mean global growth rate is also shown in **c** for
comparison (in blue). Also listed in (c) is the linear correlation coefficient (R), the mean difference and the
standard deviation of the difference of the satellite and the NOAA growth rates for 2003-2016 and for 2004-2016.





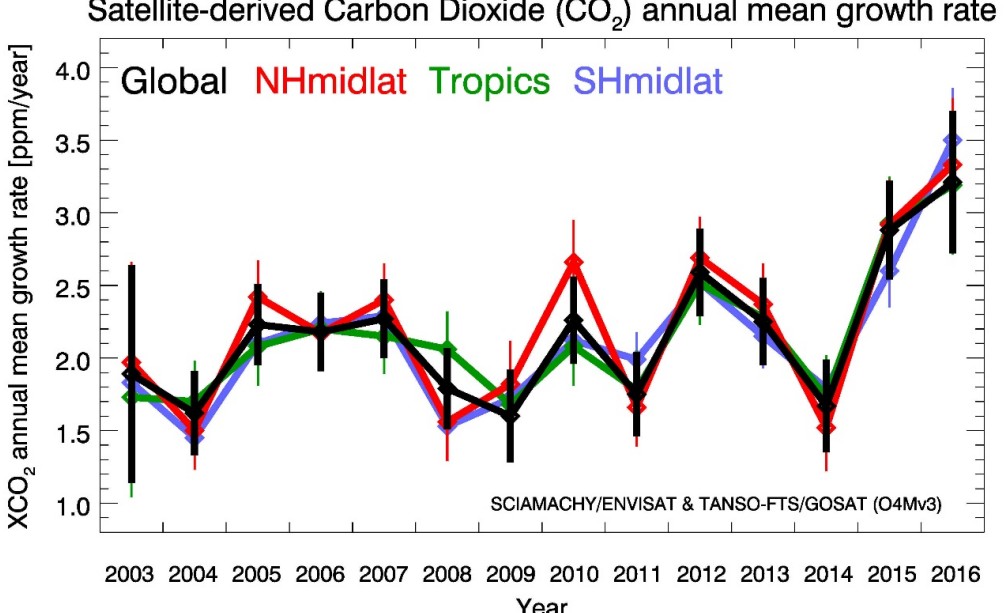

**Figure 4. Satellite-derived annual mean XCO₂ growth rates:** Global (black), Northern Hemisphere (NH) mid

latitudes ("NHmidlat" (30ºN - 60ºN), red), Tropics (30ºS - 30ºN, green), and Southern Hemisphere mid latitudes

("SHmidlat" (60ºS - 30ºS), blue). The corresponding numerical values are listed in Tab. 2.





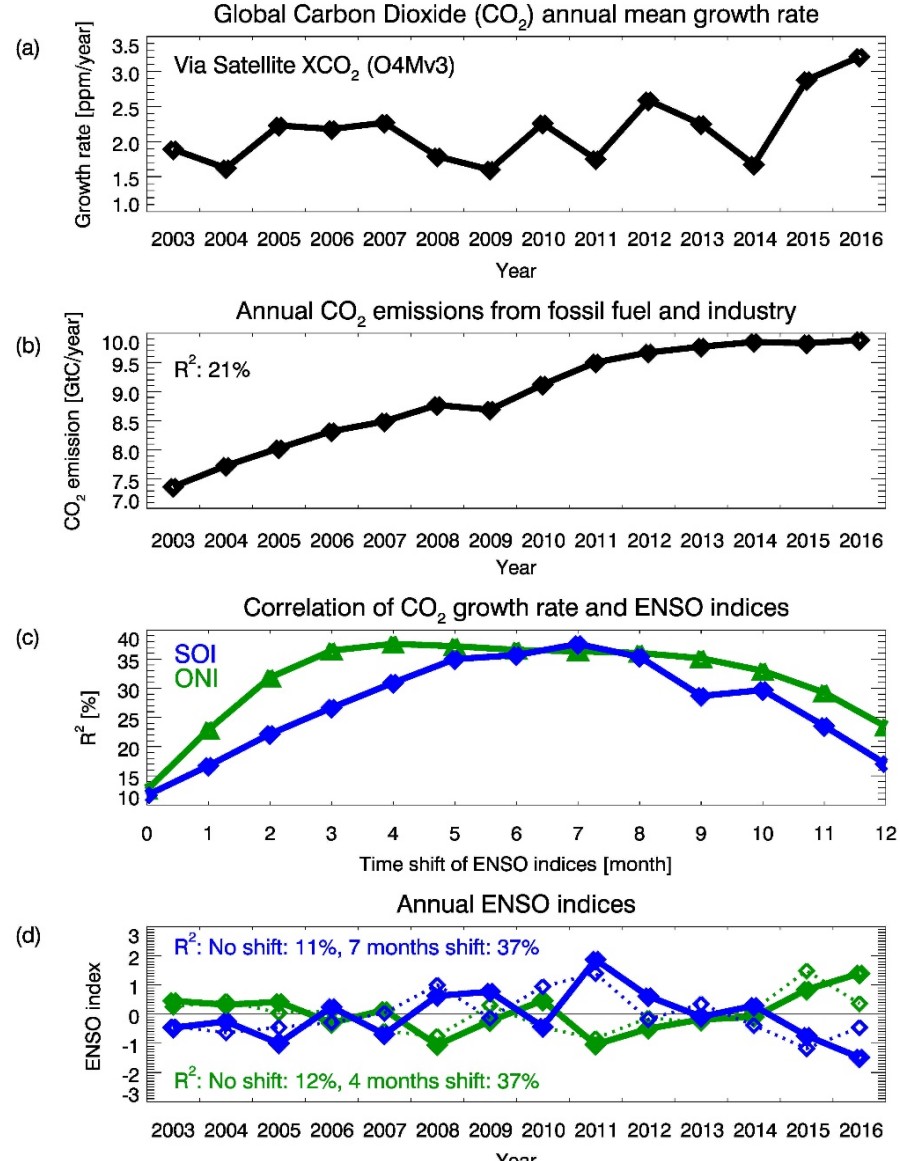

**Figure 5.** Carbon dioxide global annual mean growth rates compared with human emissions and ENSO indices.

5  (a) Satellite-derived global annual mean growth rates (same as black line in Fig. 4). (b) $CO_2$ emissions from fossil

fuel and industry (the correlation with the growth rate is $R^2 = 21\%$). (c) Correlation in terms of $R^2$ of growth rate

and annual SOI (blue curve) and ONI (green curve) as a function of time shift in months. (d) Annual SOI for no

shift (blue dotted line, $R^2 = 11\%$) and for a shift of 7 months (blue solid line, $R^2 = 37\%$) and annual ONI for no

shift (green dotted line, $R^2 = 12\%$) and for a shift of 4 months (green solid line, $R^2 = 37\%$).



**Table 1.** Satellite XCO$_2$ data products. Individual satellite sensor XCO$_2$ algorithms and corresponding Level 2 data products used for generating the EMMAv3 Level 2 (i.e., individual soundings) data product, which has been gridded to obtain the O4Mv3 Level 3 data product used in this study. GHG-CCI refers to the GHG-CCI project of ESA's Climate Change Initiative (http://www.esa-ghg-cci.org/) and C3S is the Copernicus Climate Change Service (https://climate.copernicus.eu/).

| Algorithm (Version) | Sensor | Comment | Reference |
|---|---|---|---|
| BESD (v02.01.02) | SCIAMACHY / ENVISAT | GHG-CCI / C3S product ID: CO2_SCI_BESD | Reuter et al., 2011 |
| RemoTeC (v2.3.8) | TANSO-FTS / GOSAT | GHG-CCI / C3S product ID: CO2_GOS_SRFP | Butz et al., 2011 |
| UoL-FP (v7.1) | TANSO-FTS / GOSAT | GHG-CCI / C3S product ID: CO2_GOS_OCFP | Cogan et al., 2012 |
| ACOS (v7.3.10a) | TANSO-FTS / GOSAT | NASA's GOSAT XCO$_2$ product | O'Dell et al., 2012 |
| NIES (v02) | TANSO-FTS / GOSAT | Operational GOSAT product | Yoshida et al., 2013 |



**Table 2.** Satellite-derived annual mean $XCO_2$ growth rates in ppm/year including 1-sigma uncertainty (in brackets). Abbreviations: NH is Northern Hemisphere and SH is Southern Hemisphere.

| Year | Latitude band / region | | | |
|------|--------|----------------------------|----------------------|-----------------------------|
|      | Global | NH mid-latitudes (30ºN-60ºN) | Tropics (30ºS-30ºN) | SH mid-latitudes (60ºS-30ºS) |
| 2003 | 1.89 (0.75) | 1.97 (0.69) | 1.73 (0.69) | 1.83 (0.56) |
| 2004 | 1.62 (0.29) | 1.50 (0.27) | 1.70 (0.28) | 1.45 (0.21) |
| 2005 | 2.23 (0.28) | 2.42 (0.25) | 2.08 (0.27) | 2.10 (0.19) |
| 2006 | 2.18 (0.27) | 2.17 (0.24) | 2.20 (0.26) | 2.24 (0.19) |
| 2007 | 2.27 (0.27) | 2.40 (0.25) | 2.15 (0.26) | 2.29 (0.19) |
| 2008 | 1.79 (0.28) | 1.56 (0.27) | 2.06 (0.26) | 1.53 (0.20) |
| 2009 | 1.60 (0.32) | 1.82 (0.30) | 1.68 (0.30) | 1.72 (0.21) |
| 2010 | 2.26 (0.30) | 2.66 (0.29) | 2.08 (0.27) | 2.12 (0.19) |
| 2011 | 1.75 (0.29) | 1.66 (0.27) | 1.78 (0.27) | 1.99 (0.19) |
| 2012 | 2.59 (0.30) | 2.69 (0.28) | 2.51 (0.28) | 2.52 (0.20) |
| 2013 | 2.25 (0.30) | 2.37 (0.28) | 2.28 (0.28) | 2.15 (0.22) |
| 2014 | 1.67 (0.32) | 1.52 (0.30) | 1.73 (0.29) | 1.79 (0.22) |
| 2015 | 2.88 (0.34) | 2.92 (0.31) | 2.93 (0.32) | 2.60 (0.25) |
| 2016 | 3.21 (0.49) | 3.33 (0.46) | 3.19 (0.48) | 3.50 (0.36) |

