# Peer review of "Computation and analysis of atmospheric carbon dioxide annual mean growth rates from satellite observations during 2003-2016"

_Atmospheric Chemistry and Physics, 2018_

## Referee Comment (RC1) · Anonymous Referee #1 · 7 Apr 2018

The paper describes the analysis of column-average dry-air mole fractions of CO2 observed by SCIAMACHY and GOSAT. The data being analysed represent over a decade of substantial international efforts and is an amazing accomplishment that is documented in many previous papers. The headline figures from this paper look impressive but the subsequent analysis is weak and does not add much to the main paper. Below I substantiate these comments. I recommend the paper be published but only after the major issues are addressed.

Major points

The authors will be acutely aware that it is difficult to compare NOAA ground-based

data with XCO2 data from ground-based or space-based remote sensing instruments. Columns are an integrated sum of many geographically distributed sources and sinks from a range of times that have been distributed throughout the atmosphere. Consequently, it is difficult to compare NOAA and XCO2 CO2 growth rates. Here, I am suggesting only that the authors acknowledge this as a difficulty.

The global growth rates determined by XCO2 are I believe valid and physically meaningful. However, regional growth rates (no matter how you divide the Earth) make little or no sense because of atmospheric transport that moves air from one region (e.g. zonal band indicative of midlatitudes) to another. It is tempting to interpret regional growth rates, but they are (strictly speaking) scientifically meaningless without understanding changes in atmospheric transport. By (implicitly) ignoring atmospheric transport the authors are essentially assuming that observed regional CO2 variations results exclusively from that region.

The authors' attempt at quantifying the respective role of human emissions and ENSO on CO2 growth rates is unfortunately (at least in this reviewer's opinion) a fool's errand. Our knowledge of human emissions is relatively good but still poor. Liu et al 2017 (Science) showed contrasting tropical carbon cycle responses in response to ENSO. These different responses will only complicate the correlative analysis of CO2 growth rate and ENSO indices.

Minor points

Line 6. Geological processes are only a minor sink of CO2 over decadal scales. I applaud the authors being comprehensive but this reviewer suggests a focus on the timescales that correspond to the analysis being presented.

Line 10/11. Relating GtC/yr to ppm is an undergraduate exercise that barely needs a reference let alone two.

---

## Referee Comment (RC2) · Anonymous Referee #2 · 17 Apr 2018

This manuscripts computes the CO2 growth rate from a combination of two near infrared satellite sensors over almost a decade and a half. The authors show that their estimated growth rates are in line with NOAA growth rates computed from marine boundary layer sites, and variations in the growth rate are correlated with expected mechanisms such as the ENSO cycle and anthropogenic emissions. This is all reasonable. However, I do not think that Atmospheric Chemistry and Physics is the correct journal for publishing this manuscript, because the manuscript does not present anything new about either the atmosphere or surface processes that influence the atmosphere (my comments on variation partitioning follow later). What I learned from this manuscript is that the merged XCO2 data product Obs4MIPs gives global CO2

growth rates that are reasonable, in line with other estimates, and can be correlated with known factors influencing the carbon budget. This is a perfectly fine message, but it's primarily a message about the Obs4MIPs data product, and therefore a better venue for it would be an alternative measurement- or data-focused journal such as Atmospheric Measurement Techniques or Earth System Science Data. If the authors insist on publishing this in ACP and the editor agrees, I would strongly suggest making this a technical note instead of a research article.

Regardless of where this manuscript is published, there are a few issues that I would recommend the authors address, which are as follows:

(1) I fail to see the significance of splitting the growth rate into latitude bands. The authors must be well aware that such a split, while numerically possible, is impossible to tie to any set of surface processes because of atmospheric mixing, since the interhemispheric mixing time is a year or less. What was the authors' purpose behind deriving growth rates in zonal bands?

(2) While computing the global average XCO2, did the authors account for differing surface areas at different latitudes? There is less atmospheric mass at high latitudes, and unless this is taken into account, a straight-up averaging of gridded XCO2 globally is not going to give the correct mean CO2 mole fraction, which would invalidate its link with the global flux. It's not clear from the manuscript if the authors already took care of this (the NOAA estimate includes proper weighting by surface area [Ballantyne et al, 2012]).

(3) Every El Niño is different. Some cause large changes in ocean fluxes, while others cause large changes in land fluxes, which in turn can either be ecosystem-driven or fire-driven [Sarmiento et al, 2010]. The growth rate in global CO2 is a combination of all possible factors. To try and correlate this growth rate with an ocean-only indicator like ONI or SOI is a drastic oversimplification. To then use that correlation to infer the percentage variation in the growth rate due to ENSO (as opposed to fossil fuel emissions) is even less robust. If the authors really want to dig into the factors behind CO2 variability, I would suggest some index more strongly tied to the terrestrial biosphere, such as biomass-weighted precipitation or temperature anomalies, which in turn are influenced by ENSO.

(4) SCIAMACHY sensors were degraded a few years into flight, influencing the precision of retrieved XCH4 [Frankenberg et al, 2011]. Was a similar effect seen for retrieved XCO2? If so, why doesn't that should up as larger errors bars in figure 3(c) after 2006?

References:

[1] A. P. Ballantyne, C. B. Alden, J. B. Miller, P. P. Tans, and J. W. C. White, "Increase in observed net carbon dioxide uptake by land and oceans during the past 50 years," Nature, vol. 488, no. 7409, pp. 70–72, Aug. 2012. [2] C. Frankenberg, I. Aben, P. Bergamaschi, E. J. Dlugokencky, R. van Hees, S. Houweling, P. van der Meer, R. Snel, and P. Tol, "Global column-averaged methane mixing ratios from 2003 to 2009 as derived from SCIAMACHY: Trends and variability," J. Geophys. Res., vol. 116, no. D4, p. D04302, Feb. 2011. [3] J. L. Sarmiento, M. Gloor, N. Gruber, C. Beaulieu, A. R. Jacobson, S. E. M. Fletcher, S. Pacala, and K. Rodgers, "Trends and regional distributions of land and ocean carbon sinks," Biogeosciences, vol. 7, no. 8, pp. 2351–2367, 2010.

---

## Author Response (AR1)

Michael Buchwitz on behalf of all co-authors                    5-June-2018

**Authors response to referee comments on manuscript "Computation and analysis of atmospheric carbon dioxide annual mean growth rates from satellite observations during 2003-2016" of Michael Buchwitz et al., MS No.: acp-2018-158**

This document includes our point-by-point response to the reviews, a list of all relevant changes made in the manuscript, and a marked-up manuscript version.

**Point-by-point response:**

Our point-by-point response to the reviews has been submitted via the ACP website and is already online, see:

AC1: 'Reply to Anonymous Referee #1', Michael Buchwitz, 05 Jun 2018:

https://www.atmos-chem-phys-discuss.net/acp-2018-158/acp-2018-158-AC1-print.pdf

AC2: 'Reply to Anonymous Referee #2', Michael Buchwitz, 05 Jun 2018:

https://www.atmos-chem-phys-discuss.net/acp-2018-158/acp-2018-158-AC2-print.pdf

Nevertheless, these 2 documents with our answers are attached to this document (see following pages).

**Marked-up manuscript version:**

The Marked-up manuscript version is also attached to this document (at the end).

**List of all relevant changes:**

We have aimed at carefully addressing all referee comments (see our Point-by-point response). This resulted in several major and minor modifications, which have been implemented for the revised version of our manuscript (see Marked-up manuscript version). The most relevant changes are:

- Based on a comment from one of the referees we have slightly improved our method to compute the annual mean growth rates. This resulted in slight changes of most of the numerical values as listed in the paper (see marked-up manuscript version) and required to regenerate most of the figures. These modifications, however, did not led to any major modifications so that all general conclusions are still valid.
- We have added to few additional sentences to provide better explanations as requested by the referees.
- Furthermore, we have implemented some minor text modifications at various places.

We conclude that addressing the referee comments resulted in a significantly improved version of our manuscript. We hope that the revised version of the manuscript meets the high standards of ACP.

Michael Buchwitz on behalf of all co-authors

One the following pages please see

- our response to the two referees and
- the marked-up manuscript version.

**Reply to Anonymous Referee #1**

We thank the referee for carefully reading our manuscript and for providing the critical review. In the following, we provide answers to each of the referee's comments and concerns.

Addressing these comments, concerns and questions helped us to prepare a significantly improved version of our manuscript.

General comments

C1: Referee:
The paper describes the analysis of column-average dry-air mole fractions of CO2 observed by SCIAMACHY and GOSAT. The data being analysed represent over a decade of substantial international efforts and is an amazing accomplishment that is documented in many previous papers. The headline figures from this paper look impressive but the subsequent analysis is weak and does not add much to the main paper. Below I substantiate these comments. I recommend the paper be published but only after the major issues are addressed.

Author's reply:
The primary objectives of the paper are
  (i)     to present a new global data set, which has not yet been published in a peer-reviewed journal,
  (ii)    to describe and apply a method to compute annual mean growth rates from this data set and
  (iii)   to interpret the variations of the derived annual mean growth rates.

The analysis of the annual $CO_2$ growth rates can be extended and enhanced e.g., by using appropriate and comprehensive modelling and considering additional data sets. However, we consider that our analysis is a relevant and important approach, which is independent of atmospheric model assumptions and uses available time series of data. Below we explain how we plan to address the major and minor comments as given in the review in order to improve the paper.

Major points

C2: Referee:
The authors will be acutely aware that it is difficult to compare NOAA ground-based data with XCO2 data from ground-based or space-based remote sensing instruments. Columns are an integrated sum of many geographically distributed sources and sinks from a range of times that have been distributed throughout the atmosphere. Consequently, it is difficult to compare NOAA and XCO2 CO2 growth rates. Here, I am suggesting only that the authors acknowledge this as a difficulty.

Author's reply:

We agree that annual growth rates of $CO_2$ determined from the NOAA in situ ground based measurements, which are very accurate but sparsely distributed, will not necessarily be identical with the annual growth rates computed from measurements of the dry column atmospheric mixing ratios or mole fractions of $CO_2$ measured over cloud free scenes from space.

Atmospheric $CO_2$ has different sources and sinks. Changes in the biological sources of $CO_2$, such as respiration and bacterially initiated decomposition and oxidation of organic matter, contribute significantly to changes of the $CO_2$ growth rate. Furthermore, $CO_2$ has a variety of geologic sources such as volcanic eruptions. Changes of the amount of $CO_2$ from volcanic eruption may contribute to the mean annual $CO_2$ growth rate. A small amount of CO2 is produced by the oxidation of CO initiated by OH. $CO_2$ is removed from the atmosphere by the biosphere through photosynthesis on the land and in the ocean. This accounts for the removal of around half of the $CO_2$ emitted each year. As is well known, $CO_2$ is only significantly removed chemically or photochemically at high altitude from the atmosphere by the reactions of O(1D), its short wave UV photolysis and ion-molecule reactions. In the mesosphere and thermosphere, the column of $CO_2$ is only a small component of the total column. For chemically long lived gases such as $CO_2$ differences in atmospheric ratio at any point in the atmosphere depends on the time taken for CO2 to mix. After its release or removal, which takes place primarily at the surface or in the boundary layer, the air mass with elevated or depleted $CO_2$ is transported by advection and convection and mixes into the atmosphere. This impacts on the horizontal and vertical distributions of $CO_2$. The reduction of $CO_2$ as a function of altitude enables the age of air mass to be estimated in the stratosphere, where vertical mixing is slow and varies in the range 2-8 years. In the troposphere mixing times are faster than exchange between the troposphere

and the stratosphere. It therefore cannot be expected that annual mean growth rates computed from $CO_2$ measurements at the surface are exactly identical with growth rates computed from $XCO_2$.

Nevertheless, growth-rates from NOAA are the de facto standard and therefore we think that it is very important to show comparisons with this reference data set.

In order to better acknowledge this difficulty we will add the following sentence at the end of the paragraph, where the comparison with the NOAA growth rates is presented: "Perfect agreement is not to be expected as these two growth rate time series have been obtained from $CO_2$ observations, which represent very different vertical sampling of the atmosphere (surface (NOAA) versus entire vertical column (satellite))".

C3: Referee:
The global growth rates determined by XCO2 are I believe valid and physically meaningful. However, regional growth rates (no matter how you divide the Earth) make little or no sense because of atmospheric transport that moves air from one region (e.g. zonal band indicative of midlatitudes) to another. It is tempting to interpret regional growth rates, but they are (strictly speaking) scientifically meaningless without understanding changes in atmospheric transport. By (implicitly) ignoring atmospheric transport the authors are essentially assuming that observed regional CO2 variations results exclusively from that region.

Author's reply:
We are aware of the fact that atmospheric transport cannot be ignored in this context. In our manuscript, we have not aimed at interpreting regional growth rates in terms of regional changes. In fact, we expect the growth rates to be not exactly identical but similar (taking into account the uncertainty of our growth rate) due to atmospheric transport and mixing. Therefore, we write: "Growth rate time series for several latitude bands are shown in Fig. 4. As can be seen from Fig. 4, the growth rates are similar in all latitude bands including the global results (for numerical values see Tab. 2). The reason for this is that atmospheric CO2 is long-lived and therefore well-mixed." The only figure where we aim at interpretation in terms of emissions and ENSO is Fig. 5 and here we only use the derived global growth rates. Because we use a XCO2 data set that is spatially resolved, we think that it is important to compute and discuss growth rates determined not only from globally averaged XCO2 but also from regionally averaged XCO2. This is important as this tells us something about the quality of the satellite data set and of the derived growth rates especially if one assumes that growth rates are expected to be similar for the selected regions.

To make the above argument clearer in the manuscript, we will add the following sentences in the paragraph, where we discuss Fig. 4:
"As a result of atmospheric transport and mixing, similar mean annual $CO_2$ growth rates, within their measurements error, are expected for all values derived at the different latitude bands. This behaviour is shown in Fig. 4 and is interpreted as an indication of the good quality of the satellite $XCO_2$ data product and the adequacy of the method used to compute the annual mean $CO_2$ growth rates."

C4: Referee:
The authors' attempt at quantifying the respective role of human emissions and ENSO on CO2 growth rates is unfortunately (at least in this reviewer's opinion) a fool's errand. Our knowledge of human emissions is relatively good but still poor. Liu et al 2017 (Science) showed contrasting tropical carbon cycle responses in response to ENSO. These different responses will only complicate the correlative analysis of CO2 growth rate and ENSO indices.

Author's reply:
Our approach to quantify the different roles is based on our new growth rate time series and well-established other time series. Our estimation method to quantify contributions from human emissions and ENSO is one attempt to address this aspect but we do not claim that our approach is the best possible. We think however that our approach is at least a reasonable and an important first step and we aimed at presenting our method as clearly as possible so that readers can judge to what extent they find the corresponding result useful or not.

We do not consider the task of trying to separate the impact of human emissions and that of ENSO on the mean annual $CO_2$ growth rate is a fool's errand. Rather we consider our approach is an example of an Occam's razor i.e. in explaining a thing (here: the variation of the satellite-derived growth rate), no more assumptions should be made than are necessary. Nevertheless, we agree that our growth rates may contain more information than extracted using the method applied in our paper.

The interesting work of Liu et al 2017 (Science) uses a complex earth model, constrained by a limited number of satellite observations in the tropics and other a priori knowledge, to identify different responses in the different tropical continents to the surface flux of $CO_2$ and thus carbon. Our approach to quantify the different roles of ENSO and anthropogenic fossil fuel emissions uses the reported time series of mean annual $CO_2$ growth rates and well-established time series of ENSO indices and the known estimates of anthropogenic emissions from fossil fuel combustion and industry. This approach is our attempt to address what we and others consider an important issue viz: the attribution of growth rate variations to known anthropogenic emissions from fossil fuel combustion and industry and to that from the impact of ENSO. The latter has many potential impacts on the earth system amongst which are in the tropics the creation of regions of flooding and drought, increasing fire and biomass burning and changing sea surface temperature. These effects all impact on the growth rate of $CO_2$ in different ways. However, in this study we have not tried to separate the different impacts of ENSO. Rather in this study, we attribute the importance of ENSO and the known anthropogenic fossil fuel combustion and industry sources to the observed annual growth rates. Our results are not in conflict with this scientific finding of Liu et al 2017 (Science). The use of our longer term time series of $XCO_2$ provides an opportunity when coupled with models to investigate the regional impacts of ENSO both in the tropics and the extra tropics in a separate study.

Overall, we consider that our approach is relevant, reasonable and plausible. We describe our assumptions and the derivation of the attribution clearly so that readers can reproduce the results, criticise our assumptions and make improved analyses.

Minor points

C5: Referee:
Line 6. Geological processes are only a minor sink of CO2 over decadal scales. I applaud the authors being comprehensive but this reviewer suggests a focus on the timescales that correspond to the analysis being presented.

Author's reply:
For the revised version of the manuscript we will remove the link to geological processes in the introduction.

C6: Referee:
Line 10/11. Relating GtC/yr to ppm is an undergraduate exercise that barely needs a reference let alone two.

Author's reply:
We will remove one of the two references keeping only the reference to Ballantyne et al., 2012.

**Reply to Anonymous Referee #2**

We thank the referee for carefully reading our manuscript and for providing a critical review. Below we provide point-by-point answers to each of the referee's comments and concerns. Addressing these comments, concerns and questions helped us to prepare a significantly improved version of our manuscript.

General comments

C1: Referee:
This manuscripts computes the CO2 growth rate from a combination of two near infrared satellite sensors over almost a decade and a half. The authors show that their estimated growth rates are in line with NOAA growth rates computed from marine boundary layer sites, and variations in the growth rate are correlated with expected mechanisms such as the ENSO cycle and anthropogenic emissions. This is all reasonable. However, I do not think that Atmospheric Chemistry and Physics is the correct journal for publishing this manuscript, because the manuscript does not present anything new about either the atmosphere or surface processes that influence the atmosphere (my comments on variation partitioning follow later). What I learned from this manuscript is that the merged XCO2 data product Obs4MIPs gives global CO2 growth rates that are reasonable, in line with other estimates, and can be correlated with known factors influencing the carbon budget. This is a perfectly fine message, but it's primarily a message about the Obs4MIPs data product, and therefore a better venue for it would be an alternative measurement- or data-focused journal such as Atmospheric Measurement Techniques or Earth System Science Data. If the authors insist on publishing this in ACP and the editor agrees, I would strongly suggest making this a technical note instead of a research article.

Author's reply:
We agree that the manuscript would also be appropriate for a measurement- or data-focused journal and in fact we carefully thought about this option before submission to ACP. We finally concluded that ACP is appropriate because the interpretation of the satellite-derived XCO2 data set is a focus of the manuscript.

We do not consider that our manuscript is simply a technical note. It is true that the data set and the presented analysis does not show obvious contradictions with current knowledge. However, the evidence base for current knowledge is limited to the sparse but accurate ground based measurements of the in situ mixing ratios. We present an independent data set of the dry column $CO_2$ mixing ratio or mole fraction, XCO2, and the first derived annual mean growth $CO_2$ rates using this XCO2 data set. The values are similar to those derived from the ground based in situ mixing ratio measurements. The novel nature of our manuscript is that we present
   (i)      a new global XCO2 data set covering more than a decade,
   (ii)     a method to compute annual mean growth rates from this data set,
   (iii)    a comparison with NOAA (de facto standard) growth rates, which agrees well and thereby validates both approaches and
   (iv)    an interpretation of the derived growth rates to compare the impact of ENSO on the mean annual CO2 growth rate with that from fossil fuel combustion and industry.

The analysis of XCO2 and the derived annual growth rates can be enhanced and extended, e.g., by using appropriate and probably very comprehensive modelling and considering additional data sets. This would enable regional surface fluxes to be assessed. However, we consider this as outside of the scope of the current manuscript. Nevertheless, we consider our analysis as an important step in terms of interpreting the satellite-derived growth rates. It provides independent and global knowledge about the annual mean $CO_2$ growth rate.

Our preferred option would be to publish this paper in ACP (as also supported by the other referee) but of course, it is up to the Editor to decide.

C2: Referee:
Regardless of where this manuscript is published, there are a few issues that I would recommend the authors address, which are as follows:

(1) I fail to see the significance of splitting the growth rate into latitude bands. The authors must be well aware that such a split, while numerically possible, is impossible to tie to any set of surface processes because of atmospheric mixing, since the interhemispheric mixing time is a year or less. What was the authors' purpose behind deriving growth rates in zonal bands?

Author's reply:

One motivation of the approach taken was to assess the quality of the satellite XCO2 data product and of the method developed to compute annual mean growth rates. We are aware of the fact that atmospheric transport cannot be ignored and that transport and mixing will result in similar growth rates (compared to our uncertainty) for different latitude bands. We expect the latitudinal annual CO2 growth rates and the global CO2 annual growth rates to be very similar, which is what we find. Therefore, we write: "Growth rate time series for several latitude bands are shown in Fig. 4. As can be seen from Fig. 4, the growth rates are similar in all latitude bands including the global results (for numerical values see Tab. 2). The reason for this is that atmospheric CO2 is long-lived and therefore well-mixed."

In our manuscript, we have not aimed at interpreting regional growth rates in terms of regional changes. The only figure where we aim at interpretation in terms of emissions and ENSO is Fig. 5 and here we only use the derived global growth rate. Because we use a XCO2 data set that is spatially resolved, we think that it is important to compute and discuss growth rates determined not only from globally averaged XCO2 but also from regionally averaged XCO2. This is important as this tells us something about the quality of the satellite data set and of the method used to compute growth rates. If there is a good reason, why the growth rates should be similar for all regions (see above) and if the satellite data set would not show this, then this would indicate that the satellite data or the method usedf to compute growth rates from these data would suffer from a potentially serious problem. We therefore think that it is important to compute and discuss not only growth rates computed from the global data set but also from regional sub-sets. To make this clearer we will add the following text in the paragraph, where we discuss Fig. 4:

"As a result of atmospheric transport and mixing, similar mean annual CO2 growth rates, within their measurements error, are expected for all values derived at the different latitude bands. This behaviour is shown in Fig. 4 and is interpreted as an indication of the good quality of the satellite XCO2 data product and the adequacy of the method used to compute the annual mean CO2 growth rates."

C3: Referee:
(2) While computing the global average XCO2, did the authors account for differing surface areas at different latitudes? There is less atmospheric mass at high latitudes, and unless this is taken into account, a straight-up averaging of gridded XCO2 globally is not going to give the correct mean CO2 mole fraction, which would invalidate its link with the global flux. It's not clear from the manuscript if the authors already took care of this (the NOAA estimate includes proper weighting by surface area [Ballantyne et al, 2012]).

Author's reply:
For the revised version of the manuscript, we have improved the description of our method taking the referee's comment into account. Instead of unweighted averaging, we will compute monthly XCO2 values for global or regional averages by weighting with the latitude dependent area, i.e., by weighting with the cosine of latitude. To explain this we will add these sentences: "To compute the spatially averaged XCO2 time series (shown in Fig. 2a), we first longitudinally average the XCO2 followed by the computation of the area-weighted latitudinal average of XCO2 by using the cosine of latitude as weight. We consider surface area because surface fluxes are linked to mass of CO2 (or number of CO2 molecules) rather than molecular mixing ratios or mole fractions.". Our analysis shows that this leads to minor changes of most of the numbers, figures and tables presented in our initial manuscript but it will not affect any major conclusion.

C4: Referee:
(3) Every El Niño is different. Some cause large changes in ocean fluxes, while others cause large changes in land fluxes, which in turn can either be ecosystem-driven or fire-driven [Sarmiento et al, 2010]. The growth rate in global CO2 is a combination of all possible factors. To try and correlate this growth rate with an ocean-only indicator like ONI or SOI is a drastic oversimplification. To then use that correlation to infer the percentage variation in the growth rate due to ENSO (as opposed to fossil fuel emissions) is even less robust. If the authors really want to dig into the factors behind CO2 variability, I would suggest some index more strongly tied to the terrestrial biosphere, such as biomass-weighted precipitation or temperature anomalies, which in turn are influenced by ENSO.

Author's reply:
We agree that the relationship between atmospheric CO2 growth rate variations and underlying source/sink related processes is a very complex one. We would like to contribute to a much better understanding of these links but we acknowledge that our manuscript is very limited in this respect. We have used ONI and SOI as proxies for ENSO and ENSO-related effects because these are well-established indices. Our objective was to compare the impact of

ENSO on the annual mean growth rate as compared to that of the emission from fossil fuel combustion and industry. This goal we have achieved. More detailed analysis of the impact of the individual ENSO cycles on the biosphere and the land requires comparison with complex earth system models.

C5: Referee:
(4) SCIAMACHY sensors were degraded a few years into flight, influencing the precision of retrieved XCH4 [Frankenberg et al, 2011]. Was a similar effect seen for retrieved XCO2? If so, why doesn't that should up as larger errors bars in figure 3(c) after 2006?

Author's reply:
SCIAMACHY XCH4 is retrieved from a different spectral region than XCO2. The spectral region beyond 1.6 microns in SCIAMACHY used Ge doped InGaAs detectors. These detectors were more sensitive to high energy proton bombardment. Individual detector pixels after being impacted by a high energy proton had increased noise. This effect indeed made the XCH4 error larger but the XCO2 data product is not impacted by this effect.

**The following pages show the**

**marked-up manuscript version**

[revised manuscript text omitted]

---

## Author Response (AR2)

Michael Buchwitz on behalf of all co-authors          21-August-2018

**Authors response to referee comments on revised version of manuscript "Computation and analysis of atmospheric carbon dioxide annual mean growth rates from satellite observations during 2003-2016" of Michael Buchwitz et al., MS No.: acp-2018-158**

Dear Editor,

many thanks for giving us the opportunity to respond to the referees comments and concerns and to submit a revised version of our manuscript.

Unfortunately, we have not been able to convince the referees with our initial answers and our initial revised version of our manuscript although we tried as good and carefully as possible to address all concerns.

Both referees still insist on major modifications. Because of this and because the new comments provide a better understanding on our side what exactly the concerns are, we now provide a significantly improved version of our manuscript addressing the two remaining referee comments (please see our detailed "Point-by-point response to the referees comments and concerns" below).

Implementation of the recommended changes resulted in significant modifications of our manuscript as shown below in the "List of all relevant changes" and in the "Marked-up manuscript version" attached at the end of this document.

We hope that this revised version of the manuscript is acceptable for you and for the reviewers and that it meets the high standards of ACP.

Michael Buchwitz

on behalf of all co-authors

The following pages contain the following information:

- List of all relevant changes
- Point-by-point response to the referees comments and concerns
- The marked-up manuscript version

**List of all relevant changes:**

Both reviewers criticize that we also present growth rates for latitude bands. To address this we have implemented the following modifications:

- We have removed all references to this from the abstract, from the main part and from the Conclusions section.
- Instead we have added a new Annex A where we show our results for the latitude bands (for the reason why we have not entirely removed this please see our detailed answer to the referees comments).
- The Annex contains a new figure (Fig. A2) which shows a comparison of our growth rate uncertainties with the difference between two NOAA annual mean growth rate time series (Mauna Loa – Global) to support that "we expect similar annual mean $CO_2$ growth rates, i.e., agreement within measurement error, for the different latitude bands and globally". We have added this in response to the referees concern.
- As a consequence of this the corresponding figure (now Fig. A1) and table (now Tab. A1) has been been moved from the main part to the Annex and one figure has been removed.
- The figure now shown as Fig. 3 has been improved by adding error bars to the time series shown in Fig. 3a.

Both reviewers also provide critical comments related to our growth rate variance analysis. To address this we are now

- providing in Sect. 4 an additional introduction paragraph which includes and explicit formulation of the question we are answering (in Sect. 4), how we answer it and what our main assumptions are. We have added this to address the concern that is was not clear "what we are doing and why" and what the "new knowledge" is (note that in our point-by-point response to the referees comments we now also provide a list of what we consider the most relevant "new knowledge" provided by our paper). In this context, we have also added two additional references (Betts et al., 2016, and Kim et al., 2016). We hope that all is much clearer now by explicitly formulating a question (which we think is an important one) and the corresponding answer obtained using a transparent and well-explained method.

Furthermore, we have implemented minor text changes at various places to further improve the manuscript.

**Reply to Anonymous Referee #1 comments on revised manuscript**

In the following, we provide answers to each of the referee's comments and concerns.

General:

Referee C1:
My two main criticisms of this paper were their use of: 1) regional atmospheric growth rates and 2) a simple statistical model to attribute changes in atmospheric CO2 to human emissions and ENSO. Neither comment is addressed well by the authors.

Author's reply:
In our reply to your initial comments and concerns, we aimed at providing clear and appropriate answers and modified our manuscript accordingly. Too bad that we failed to explain our arguments good enough and/or that our arguments do not convince you. We try to do better this time and have also implemented major manuscript modifications. Please see below our feedback to your remaining / new comments. Based on your comments we have generated another (second) revised version of our manuscript.

Results for latitude bands:

Referee C2:
Their response to my first comment is limited to two additional statements neither of which I fully understand: a) "Growth rate time series for several latitude bands are shown in Fig. 4. As can be seen from Fig. 4, the growth rates are similar in all latitude bands including the global results (for numerical values see Tab. 2). The reason for this is that atmospheric CO2 is long-lived and therefore well-mixed." b) "As a result of atmospheric transport and mixing, similar mean annual CO2 growth rates, within their measurements error, are expected for all values derived at the different latitude bands. This behaviour is shown in Fig. 4 and is interpreted as an indication of the good quality of the satellite XCO2 data product and the adequacy of the method used to compute the annual mean CO2 growth rates." Atmospheric CO2 is well mixed but the authors are trying to determine the small, annual changes that sit on the growing well-mixed background. Atmospheric CO2 has an interhemispheric gradient, which is determined by hemispheric differences in emissions but also by the ~1 year mean interhemispheric crossing time. It can take weeks-months for signals to be transported from the tropics to midlatitudes so a seasonal change in one latitude band in year one may straddle growth rates in that year and year two. I also don't follow their argument about using regional growth rates to comment on the quality of the CO2 data.

Author's reply:
In our response to your earlier comments on this topic, we tried to explain in detail why we have applied our method also to latitude bands (and not only to the global data set). Thanks to your new comment (shown above) we now understand your concern better.

In our manuscript, we focus on the gobal results (comparison with NOAA growth rates, time series correlation analysis with emissions and ENSO indices) but we think that it is important to also show results for latitude bands. As explained, we expect similar growth rates for all latitude bands and we show that this is what we find. Our findings therefore agree with our assumptions (our knowledge) and we interpret this (as written in our manuscript) as a confirmation of the good quality of the satellite data and of the method to derive growth rates from these data as we do not get "strange" growth rates for certain latitude bands which would indicate a potential problem.

Unfortunately, we neither managed to convince you nor the other referee. Therefore, we now decided to remove all results related to latitude bands from the abstract, from the main part and from the Conclusions section but to show the results related to latitude bands instead in a new dedicated Annex A. We also discussed to entirely remove all results related to latitude bands from the manuscript but for the reasons explained (see above and below) we think that these are important results that should be shown in the paper and we think that moving this into an Annex is a good compromise. We are now also better addressing your concerns as stated in your comment by adding an additional figure, which shows an estimate of the expected latitudinal difference (for details see below).

This results in text modifications at various places including the Abstract, main text and Conclusions section and it results in (i) moving Fig. 4 and Tab. 2 to the new Annex A, (ii) the removal of Fig. 2 and (iii) adding a new Figure A2.

Here the text we plan to show in the new Annex A:

"Growth rate time series have also been computed for several latitude bands as shown in Fig. A1. As can be seen, the growth rates agree within their 1-sigma uncertainty range in all latitude bands including the global results (for numerical values see Tab. A1).
The reason for this is that atmospheric $CO_2$ is long-lived and therefore well-mixed. Because of this we expect similar annual mean $CO_2$ growth rates, i.e., agreement within measurement error, for the different latitude bands and globally. Identical growth rates are not expected due to differences in the sources and sinks and the time needed for transport and mixing. The expectation of similar growth rates is corroborated by Fig. A2, which shows a comparison of the uncertainty of the satellite-derived growth rates (red bars) with the difference of two annual mean $CO_2$ growth rate time series from NOAA, namely the time series from Mauna Loa, Hawaii, and the global time series obtained from globally averaged marine surface data (both obtained from https://www.esrl.noaa.gov/gmd/ccgg/trends/gr.html). As shown in Fig. A2, the uncertainty of the satellite data is similar (mean value: 0.34 ppm/year) as the difference between the two NOAA time series (standard deviation: 0.21 ppm/year). We acknowledge that the maximum difference between any two latitude bands may be somewhat larger than the difference between the two NOAA time series shown in Fig. A2, but it is assumed that the difference shown in Fig. A2 is at least a reasonable approximation.
The agreement shown in Fig. A1 is interpreted as an indication of the good quality of the satellite $XCO_2$ data product and of the adequacy of the method used to compute the annual mean $CO_2$ growth rates because we do not find "strange values" in certain latitude bands or certain years, which would be an indication for a potential problem."

Time series variance analysis:

Referee C3:
Their response to my second comment is a bit odd in my opinion. I agree that attempting to attribute observed changes in atmospheric CO2 to human emissions and ENSO is of great importance. However, my original comment said that the method they used to attribute human emissions and ENSO was a fool's errand. I made this comment for a number of reasons:
* the spatial and temporal distributions of human emissions and the manifold responses of the land biosphere to regional weather patterns are not necessarily distinct.
* is there a disprovable result reported by this correlation analysis? The approach/results are so amorphous and ill defined that I find it hard to understand what new knowledge I have gained from this analysis.
* if, as the authors claim, their analysis are not in conflict with Liu et al then they should show it.

Author's reply:

Concerning "spatial and temporal distributions of human emissions and land biosphere responses":
They may not be perfectly distinct but we assume that they are to a large degree distinct. After all most human emissions from fossil fuel burning do not take place where, for example, most of the land biosphere is located. The spatial distribution should not be a major issue for our analysis as we focus on global averages (and for latitude bands please see above). Concerning temporal distributions and land biosphere responses we address this (at least to some extent) by our time lag analysis and our time lag analysis results agree well with previous research as shown in our manuscript. Furthermore, we also take the temporal correlation between human emissions and the (time shifted) ENSO indices into account.

Concerning "Is there a disprovable result reported by this correlation analysis?":
We have formulated all our conclusions carefully in order not to claim something that is not supported by our analysis. Our main conclusion is (see Conclusions section): "This analysis shows that the ENSO impact on $CO_2$ growth rate variations dominates over that of human emissions throughout the period 2003-2016 but in particular in the second half of this period, i.e., during 2010-2016". This is a disprovable result as, in principle, someone may show that this is wrong. In our analysis, we assume that the growth rate variation in the investigated time period is dominated by human emissions and ENSO. We furthermore assume that ENSO is well described by the used ENSO indices. If these assumptions are not valid, then our conclusions may be wrong. However, we also consider the uncertainties we are reporting. Therefore, in more quantitative terms, we conclude: "We estimate the probability that the impact of ENSO on the variability is larger than the impact of human emissions to be 63% for the time period 2003-2016. If the time period is restricted to 2010-2016 this probability increases to 94%". These statements are based on Monte Carlo simulations taking into account the uncertainties of the growth rates. The percentages show that we are quite sure that our findings are robust for the period 2010-2016 but that we are less sure for the period 2003-2016.
In the new revised version we will explain more explicitly what our main assumption are by adding this new paragraph at the beginning of Sect. 4: 'It is well known that changes of the growth rate of atmospheric $CO_2$ have anthropogenic and natural causes (e.g., Jones et al., 2001; Betts et al., 2016; Kim et al., 2016; Liu et al., 2017; Chylek et al., 2018). In this section we are aiming at answering the following question: "Assuming that the variability of the $CO_2$ growth rate is dominated by ENSO and by human emissions, which of the two considered causes dominates the growth rate variability given the satellite-derived growth rates and their uncertainty?". To answer this question we are using a simple linear statistical model and time series of human emissions and two ENSO indices assuming that these indices are appropriate proxies for ENSO related effects in the context of providing a reliable answer.'. Concerning "new knowledge" please see the list we provide at the end of our response.

This paragraph contains two additional references, which we added to our manuscript:

Betts et al., 2016: Betts, R. A., Jones, C. D., Knight, J. R., Keeling, R. F., and Kennedy, J. J., El Niño and a record $CO_2$ rise, Nature Climate Change, vol. 6, 806–810, https://www.nature.com/articles/nclimate3063.pdf , 2016.

Kim et al., 2016: Kim, J.-S., Kug, J.-S., Yoon, J.-H., Jeong, S.-J., Increased Atmospheric $CO_2$ Growth Rate during El Niño Driven by Reduced Terrestrial Productivity in the CMIP5 ESMs, Journal of Climate, 8783-8805, 29. 10.1175/JCLI-D-14-00672.1, 2016.

Concerning "amorphous and ill defined":

We disagree that our approach is "amorphous and ill defined". The opposite is true: We use a very simple and well-defined method based on well established other data sets (published and publicly available annual $CO_2$ emissions and time series of ENSO indices) and we explain everything clearly so that it can be easily reproduced by others. We agree, however, that the problem is a complex one. Here we aim at answering only one specific question (as already explained above, this question is now given explicitly in our manuscript at the beginning of Sect. 4): "Assuming that the variability of the $CO_2$ growth rate is dominated by ENSO and by human emissions, which of the two considered causes is the dominating one given the satellite-derived growth rates and their uncertainty?" We clearly explained this method and its results (i.e., the answer to the question) and this is new knowledge presented in our manuscript (for the complete list of "new knowledge" please see below). So we answered one question (which we think is an interesting one) but we were not aiming at answering all questions related the impact of ENSO and anthropogenic emissions on $CO_2$ growth rates.

Concerning Liu et al:

In our paper we cite the Liu et al., 2017, Science paper only in the context of the discussion of the large 2015/2016 growth rates: "As can also be seen from Fig. 2c, the largest growth rates are approximately 3 ppm/year during 2015 and 2016. These record large growth rates (Peters et al., 2017) are attributed to the consequences of the strong 2015/2016 El Niño event, which produced large $CO_2$ emissions from fires and enhanced net biospheric respiration in the tropics relative to normal conditions (Heymann et al., 2017; Liu et al., 2017)". In our paper, we do not claim anything that goes beyond this. We only refer to the Liu et al. and Heymann et al. papers as they provide relevant information in the context of the discussion of the growth rates. In particular we have not identified anything that points to a (potential) conflict.

However, in our response to your initial comments we wrote: "The interesting work of Liu et al 2017 (Science) uses a complex earth model, constrained by a limited number of satellite observations in the tropics and other a priori knowledge, to identify different responses in the different tropical continents to the surface flux of $CO_2$ and thus carbon. Our approach to quantify the different roles of ENSO and anthropogenic fossil fuel emissions uses the reported time series of mean annual $CO_2$ growth rates and well-established time series of ENSO indices and the known estimates of anthropogenic emissions from fossil fuel combustion and industry. This approach is our attempt to address what we and others consider an important issue viz: the attribution of growth rate variations to known anthropogenic emissions from fossil fuel combustion and industry and to that from the impact of ENSO. The latter has many potential impacts on the earth system amongst which are in the tropics the creation of regions of flooding and drought, increasing fire and biomass burning and changing sea surface temperature. These effects all impact on the growth rate of $CO_2$ in different ways. However, in this study we have not tried to separate the different impacts of ENSO. Rather in this study, we attribute the importance of ENSO and the known anthropogenic fossil fuel combustion and industry sources to the observed annual growth rates. Our results are not in conflict with the scientific finding of Liu et al 2017 (Science). The use of our longer term time series of $XCO_2$ provides an opportunity when coupled with models to investigate the regional impacts of ENSO both in the tropics and the extra tropics in a separate study. Overall, we consider that our approach is relevant, reasonable and plausible. We describe our assumptions and the derivation of the attribution clearly so that readers can reproduce the results, criticise our assumptions and make improved analyses.". This answer contains the sentence "Our results are not in conflict with the scientific finding of Liu et al 2017 (Science)." We have written this because of the explanation as given above. But perhaps this sentence is too strong / misleading. What we mean is that we have not identified any aspect where we are in conflict with the findings of Liu et al. (and nothing in this direction is mentioned in our manuscript). We should have explained this better in our response to your initial comments and we apologize for not having formulated this clear enough.

Concerning "new knowledge":

In our manuscript, we present the following new knowledge:

- We present a new global total column $CO_2$ ("$XCO_2$") data set (based on satellite data) covering 14 years
- We present a new method to compute annual mean $XCO_2$ growth rates from this data set
- We present a new annual mean $CO_2$ growth rate time series (covering the entire atmosphere, not only near-surface $CO_2$) including a comparison with growth rates from NOAA based on surface $CO_2$ observations; we find agreement within the reported uncertainty ranges and therefore consider our growth rates to be validated
- We present an answer to the question "Assuming that the variability of the $CO_2$ growth rate is dominated by ENSO and by human emissions, which of the two considered causes dominates the growth rate variability given the satellite-derived growth rates and their uncertainty?" To answer this question we used a statistical analysis method, which we clearly explain. Our answer is given in the Conclusions section: "Our analysis also shows that the ENSO impact on $CO_2$ growth rate variations dominates over that of human emissions throughout the period 2003-2016 (14 years) but in particular during the period 2010-2016 (second half of the investigated time period) due to strong La Niña and El Niño events. We estimate the probability that the impact of ENSO on the variability is larger than the impact of human emissions to be 63% for the time period 2003-2016. If the time period is restricted to 2010-2016 this probability increases to 94%."

**Reply to Anonymous Referee #2 comments on revised manuscript**

In the following, we provide answers to each of the referee's comments and concerns. Based on these comments we have generated another (second) revised version of our manuscript.

Referee C1:
The authors have responded to most of my technical comments carefully.

Author's reply:
This is good to know. In fact, we tried to address all comments as good and carefully as possible.

Referee C2:
However, I do not see that they changed much scientifically. E.g., both referees pointed out that a zonal partitioning of the growth rate has little meaning for CO2, yet it's still there, this time with a small disclaimer.

Author's reply:
Unfortunately, you are not referring to our detailed justification as provided in our response to your initial review and why our explanations do not convince you.

In our manuscript, we focus on the gobal results (comparison with NOAA growth rates, time series correlation analysis with emissions and ENSO indices) but we think that it is important to also show results for latitude bands. As explained, we expect similar growth rates for all latitude bands and we show that this is what we find. Our findings therefore agree with our assumptions (our knowledge) and we interpret this (as written in our manuscript) as a confirmation of the good quality of the satellite data and of the method to derive growth rates from these data as we do not get "strange" growth rates for certain latitude bands which would indicate a potential problem.

Unfortunately, we neither managed to convince you nor the other referee. Therefore, we now decided to remove all results related to latitude bands from the abstract, from the main part and from the Conclusions section but to show the results related to latitude bands instead in a new dedicated Annex A. We also discussed to entirely remove all results related to latitude bands from the manuscript but for the reasons explained (see above and below) we think that these are important results that should be shown in the paper and we decide that moving this into an Annex would be a good compromise. We are now also better addressing your concerns as stated in your comment by adding an additional figure, which shows an estimate of the expected latitudinal difference (for details see below).

This results in text modifications at various places including the Abstract, main text and Conclusions section and it results in (i) moving Fig. 4 and Tab. 2 to the new Annex A, (ii) the removal of Fig. 2 and (iii) adding a new Figure A2.

Here the text we plan to show in the new Annex A:

"Growth rate time series have also been computed for several latitude bands as shown in Fig. A1. As can be seen, the growth rates agree within their 1-sigma uncertainty range in all latitude bands including the global results (for numerical values see Tab. A1).
The reason for this is that atmospheric $CO_2$ is long-lived and therefore well-mixed. Because of this we expect similar annual mean $CO_2$ growth rates, i.e., agreement within measurement error, for the different latitude bands and globally. Identical growth rates are not expected due to differences in the sources and sinks and the time needed for transport and mixing. The expectation of similar growth rates is corroborated by Fig. A2, which shows a comparison of the uncertainty of the satellite-derived growth rates (red bars) with the difference of two annual mean $CO_2$ growth rate time series from NOAA, namely the time series from Mauna Loa, Hawaii, and the global time series obtained from globally averaged

marine surface data (both obtained from https://www.esrl.noaa.gov/gmd/ccgg/trends/gr.html). As shown in Fig. A2, the uncertainty of the satellite data is similar (mean value: 0.34 ppm/year) as the difference between the two NOAA time series (standard deviation: 0.21 ppm/year). We acknowledge that the maximum difference between any two latitude bands may be somewhat larger than the difference between the two NOAA time series shown in Fig. A2, but it is assumed that the difference shown in Fig. A2 is at least a reasonable approximation.

The agreement shown in Fig. A1 is interpreted as an indication of the good quality of the satellite $XCO_2$ data product and of the adequacy of the method used to compute the annual mean $CO_2$ growth rates because we do not find "strange values" in certain latitude bands or certain years, which would be an indication for a potential problem."

Referee C3:
Moreover, I remain unconvinced that the work is significant enough to qualify for a standalone ACP publication. Typically the analysis presented in this paper would be a small part of a larger paper, say a paper on a source-sink inversion of SCIAMACHY XCO2 or the validation of the XCO2 retrieval algorithm. However, a quantification of the global growth rate (zonal bands, as I said, mean very little) and a comparison to NOAA's MBL growth rate, in my opinion, does not qualify as a solid standalone publication. On this point, I am afraid, the authors and I may never agree.

Author's reply:
As already explained on our detailed response to your initial comments, we do not agree with this. As explained, we think that our manuscript is appropriate for ACP because of the topic and because of the new results, we are presenting.

In our manuscript, we present the following new knowledge:
In our manuscript, we present the following new knowledge:
- We present a new global total column $CO_2$ ("$XCO_2$") data set (based on satellite data) covering 14 years
- We present a new method to compute annual mean $XCO_2$ growth rates from this data set
- We present a new annual mean $CO_2$ growth rate time series (covering the entire atmosphere, not only near-surface $CO_2$) including a comparison with growth rates from NOAA based on surface $CO_2$ observations; we find agreement within the reported uncertainty ranges and therefore consider our growth rates to be validated
- We present an answer to the question "Assuming that the variability of the $CO_2$ growth rate is dominated by ENSO and by human emissions, which of the two considered causes dominates the growth rate variability given the satellite-derived growth rates and their uncertainty?" To answer this question we used a statistical analysis method, which we clearly explain. Our answer is given in the Conclusions section: "
[revised manuscript text omitted]

| 2003 | 1.66 (0.76) | 1.99 (0.72) | 1.54 (0.74) | 1.77 (0.62) |
| 2004 | 1.59 (0.30) | 1.52 (0.29) | 1.71 (0.29) | 1.30 (0.23) |
| 2005 | 2.16 (0.28) | 2.51 (0.26) | 1.99 (0.28) | 2.17 (0.22) |
| 2006 | 2.21 (0.27) | 2.13 (0.25) | 2.22 (0.27) | 2.33 (0.21) |
| 2007 | 2.26 (0.27) | 2.33 (0.25) | 2.20 (0.26) | 2.34 (0.21) |
| 2008 | 1.67 (0.29) | 1.60 (0.27) | 1.81 (0.28) | 1.41 (0.20) |
| 2009 | 1.77 (0.30) | 1.75 (0.30) | 1.86 (0.28) | 1.70 (0.21) |
| 2010 | 2.22 (0.29) | 2.67 (0.29) | 2.08 (0.27) | 2.14 (0.20) |
| 2011 | 1.86 (0.28) | 1.69 (0.27) | 1.86 (0.27) | 2.19 (0.19) |
| 2012 | 2.46 (0.29) | 2.64 (0.28) | 2.44 (0.27) | 2.38 (0.21) |
| 2013 | 2.27 (0.30) | 2.38 (0.28) | 2.27 (0.28) | 2.10 (0.22) |
| 2014 | 1.74 (0.31) | 1.53 (0.30) | 1.80 (0.29) | 1.84 (0.23) |
| 2015 | 2.89 (0.34) | 2.89 (0.31) | 2.97 (0.32) | 2.54 (0.25) |
| 2016 | 3.23 (0.50) | 3.28 (0.46) | 3.23 (0.48) | 3.41 (0.36) |

---

## Author Response (AR3)

**Authors response to referee comments on revised version of manuscript "Computation and analysis of atmospheric carbon dioxide annual mean growth rates from satellite observations during 2003-2016" of Michael Buchwitz et al., MS No.: acp-2018-158**

Dear Editor,

many thanks for giving us the opportunity to respond to the comments and concerns of the two new referees and to submit a revised version of our manuscript.

We provide an improved version of our manuscript addressing the comments of the referees as good as possible. Please see our detailed "Point-by-point response to the comments and concerns of the referees" below.

Implementation of the recommended changes resulted in modifications of our manuscript as shown below in the "List of all relevant changes" and in the "Marked-up manuscript version" attached at the end of this document.

We hope that this revised version of the manuscript is acceptable for you / ACP and that it meets the high standards of ACP.

   Michael Buchwitz

   on behalf of all co-authors

The following pages contain the following information:

- List of all relevant changes
- Point-by-point response to the comments and concerns of the referees
- The marked-up manuscript version

**List of all relevant changes:**

To consider the comments from Referee #4 we have implemented the following modifications:

Abstract:

We added two sentences (from the conclusions section) to provide more detailed conclusions in the abstract as requested by the referee: "Our analysis shows that the ENSO impact on $CO_2$ growth rate variations dominates over that of human emissions throughout the period 2003-2016 but in particular during the period 2010-2016 due to strong La Niña and El Niño events. Using the derived growth rates and their uncertainty, we estimate the probability that the impact of ENSO on the variability is larger than the impact of human emissions to be 63% for the time period 2003-2016. If the time period is restricted to 2010-2016 this probability increases to 94%.".

Section 3:

We provide more details concerning the differences and similarities of our method and the NOAA method: We removed the sentence "We adopt this definition" in the 1$^{st}$ paragraph and modified the corresponding paragraph near the end of Sect. 3. Modified text: "Perfect agreement is not to be expected as these two growth rate time series have been obtained from $CO_2$ observations, which represent very different vertical sampling of the atmosphere (surface (NOAA) versus entire vertical column (satellite)) (see Fig. A3b in Annex A for a comparison of $XCO_2$ and surface $CO_2$ growth rates obtained using a global re-analysis $CO_2$ data product). Perfect agreement is also not to be expected because we use different time periods for the computation of the annual growth rates compared to NOAA (see Fig. A3c in Annex A for a comparison of two different methods to compute annual $XCO_2$ growth rates).".

We also added 2 sentences to consider the referee's comment on our uncertainty estimates. Text added: "We aimed at providing realistic error estimates but we acknowledge that our uncertainty estimates are not based on full error propagation, which would be difficult especially due to unknown or not well enough known systematic errors and error correlations. The reported uncertainty estimates should therefore be interpreted as error indications rather than fully rigorous error estimates.".

To compare growth rates computed from $XCO_2$ and surface $CO_2$ we have downloaded and analysed a (large) multi-year $CO_2$ data set and used it to generate a new figure incl. discussion. The reference to this new figure is given in Sect. 3 and the new figure has been added in the Annex as Fig. A3. This required to also add a new reference (Chevallier, 2018).

Section 4:

We have added these sentences at the beginning of the 3rd paragraph: "Figure 3 shows that the anthropogenic emission variability is mostly linked to a trend whereas the El Niño signal is variable on much shorter time scales. Thus, the relative impact of the anthropogenic and natural contributions depends on the length of the time series. The shorter the time series, the smaller the anthropogenic variability is. It is therefore expected that the natural contribution to the variability of the growth rate gets larger for a shorter time series.".

Acknowledgements:

We have added this at the end: "The CAMS $CO_2$ re-analysis data set has been obtained http://apps.ecmwf.int/datasets/data/cams-ghg-inversions/ (access: 13-Nov-2018). Finally, we would like to thank four anonymous referees for helpful comments.".

To consider the specific comment of Referee #3 we have implemented the following modification:

Section 3, 1st paragraph:

We have added "(e.g., a latitude band, see Annex A, Fig. A1)" as an additional explanation to item (i).

**Reply to Anonymous Referee #4 (Report #2) comments on revised manuscript**

In the following, we provide answers to each of the referee's comments and concerns. Based on these comments we have generated another (third) revised version of our manuscript.

General:

Referee C1:
This paper presents satellite observation data of the atmospheric Carbon Dioxide, derives an ad-hoc method to estimate a growth rate, and analyses the variability of the annual growth rate with respect to the anthropogenic (human emissions) and natural (ENSO) contributions. This paper already went through two rounds of exchanges between the authors and two reviewers. I understand that the two reviewers have given-up and I have been asked to provide a "final" review on the revised version of the paper. The two reviewers had rather consistent comments. One comment focused on the analysis of the growth rate per latitude band and its interpretation. This analysis has now been removed from the main part of the paper and moved into an Annex. One can therefore state that this concern is resolved. The other comment was on the poor significance of the paper. Although I can agree with the fact that the scientific content of the paper is limited, I do see a large fraction of the papers published with similar or even lower scientific content. The current trend in the scientific literature is unfortunate, but this particular paper should not be a first target in this context. My opinion is then that the version of the paper that I have been asked to review can be published in ACP

Author's reply:
Many thanks for reviewing our manuscript.

Specific comments:

Referee C2:
I nevertheless take this opportunity to point a few things that apparently were not mentioned by the two original reviewers

Author's reply:
Your additional comments are all very good and as shown below we aimed at addressing them as good as possible for the revised version of the manuscript.

Referee C3:
The last sentence of the abstract states what has been done in the paper concerning the variability of the growth rate but does not provide the conclusion that can be fairly easily stated. I recommend that the conclusion is given in the abstract

Author's reply:
We have added these sentences (from the conclusions section) at the end of the abstract: "Our analysis shows that the ENSO impact on $CO_2$ growth rate variations dominates over that of human emissions throughout the period 2003-2016 but in particular during the period 2010-2016 due to strong La Niña and El Niño events. Using the derived growth rates and their uncertainty, we estimate the probability that the impact of ENSO on the variability is larger than the impact of human emissions to be 63% for the time period 2003-2016. If the time period is restricted to 2010-2016 this probability increases to 94%.".

Referee C4:
Page 4, line 20-28, the authors describe how they compute an uncertainty on the growth rate. The method is far from rigorous and there is absolutely no argument why the uncertainty should be estimated as the average of three terms. It should be made clear that the value that is derived is only an indicator, but in no mean a proper uncertainty estimate.

Author's reply:
In our manuscript we explain in detail why we have chosen this approach. To take your comment into account we added these sentences at the end of the paragraph, where our method is explained: "We aimed at providing realistic error estimates but we acknowledge that our uncertainty estimates are not based on full error propagation, which would be difficult especially due to unknown or not well enough known systematic errors and error correlations. The reported uncertainty estimates should therefore be interpreted as error indications rather than fully rigorous error estimates.".

Referee C5:
The annual growth rate that is computed is an annual average of monthly estimates that are themselves computed from the differences between the XCO2 averages over a year. Thus, the 2010 growth rate (for instance) involves measurements from July 2009 to July 2011. Conversely, the NOAA growth rate (based on surface measurements) that is used for an evaluation from they production uses measurements of January and December 2010. The time periods are different and this should be made very clear.

Author's reply:
Based on your comments we have removed the sentence "We adopt this definition" and modified the sentences at the end of the paragraph, where the comparison with NOAA is discussed. The modified text is the following: "Perfect agreement is not to be expected as these two growth rate time series have been obtained from $CO_2$ observations, which represent very different vertical sampling of the atmosphere (surface (NOAA) versus entire vertical column (satellite)) (see Fig. A3b in Annex A for a comparison of $XCO_2$ and surface $CO_2$ growth rates obtained using a global re-analysis $CO_2$ data product). Perfect agreement is also not to be expected because we use different time periods for the computation of the annual growth rates compared to NOAA (see Fig. A3c in Annex A for a comparison of two different methods to compute annual $XCO_2$ growth rates).".

Referee C6:
When doing the comparison of the growth rate based on surface and satellite data (comment above) (page 5 line 9 and Figure 2), the author argues that the difference is linked to the difference vertical sampling (line 9). To substantiate that statement, they could use a the results from a global transport model and compare the annual growth rate at the surface from that of the full column. I am sure that several of the authors have such simulations available so that this simple check would be easy to achieve.

Author's reply:
To address this comment we have downloaded a multi-year $CO_2$ re-analysis data set and used it to compute and compare annual mean growth rates from $XCO_2$ and surface $CO_2$. The new results are presented and discussed in Annexe A, which now includes a new Fig. 3A. Here the corresponding new text in Annex A: "Figure A3 shows a comparison of $XCO_2$ and surface $CO_2$ annual growth rates as computed from a Copernicus Atmosphere Monitoring Service (CAMS) global re-analysis $CO_2$ data set (Chevallier, 2018). This CAMS atmospheric $CO_2$ data set does not (in contrast to satellite data) suffer from data gaps and measurement noise. Therefore, the annual growth rate can simply be computed from the difference of the ($XCO_2$ or surface $CO_2$) values at the end of a year and the beginning of that year ("method M1"). Figure A3b confirms that growth rates computed (using method M1) from $XCO_2$ and from surface $CO_2$ are very similar but not exactly identical. Figure A3c shows that the satellite method ("M2") described in this publications provides annual $XCO_2$ growth rates, which are very similar to those obtained with the M1 method.".

Corresponding new reference:

Chevallier, F., Validation report for the inverted $CO_2$ fluxes, v17r1, Technical Report Copernicus Atmosphere Monitoring Service (CAMS), version 1.0 (06/07/2018), available from CAMS website (https://atmosphere.copernicus.eu/sites/default/files/2018-10/CAMS73_2015SC3_D73.1.4.2-1979-2017-v1_201807_v1-1.pdf), 2018.

Referee C7:

Figure 3 very clearly shows that the anthropogenic emission variability is mostly linked to a trend whereas the El Nino signal is variable on much shorter time scales. Thus, the relative impacts of the anthropogenic and natural contributions will very much depend on the length of the time series. The shorter the time series, the smaller the anthropogenic variability is. It is then very much expected that the natural contribution to the variability of the growth rate gets larger for a shorter time series. This should be made clear in the manuscript and conclusions that are somewhat misleading in this respect

Author's reply:

To address this comment we have added this sentence at the beginning of the paragraph where we separate and quantify the anthropogenic and ENSO contributions: "Figure 3 shows that the anthropogenic emission variability is mostly linked to a trend whereas the El Niño signal is variable on much shorter time scales. Thus, the relative impact of the anthropogenic and natural contributions depends on the length of the time series. The shorter the time series, the smaller the anthropogenic variability is. It is therefore expected that the natural contribution to the variability of the growth rate gets larger for a shorter time series.".

**Reply to Anonymous Referee #3 (Report #1) comments on revised manuscript**

In the following, we provide answers to each of the referee's comments and concerns.

General:

Referee C1:
Review of "Computation and analysis of atmospheric carbon dioxide annual mean growth rates from satellite observations during 2003-2016" by Buchwitz et al.

The authors presented a new Level 3 XCO2 product, based on data from SCIAMACHY and GOSAT, and examined the atmospheric growth rate of CO2 captured by these data. They showed that the annual mean CO2 growth rate estimated from the XCO2 data is consistent with that estimated from the NOAA surface CO2 data. They also did an analysis to determine the relative contributions of ENSO and anthropogenic CO2 emissions to variations in the annual mean CO2 growth rate. The new Level 3 data will be a useful product for the community since working with Level 2 data can be challenging. The fact that the XCO2-based CO2 growth rate is consistent with that estimated from the surface data is reassuring. However, I cannot recommend the manuscript for publication in ACP. I do not believe that the manuscript contains sufficiently new scientific results to warrant publication in ACP.

In responding to the previous reviews, the authors described what new knowledge is contained in the manuscript. They stated that:
• "We present a new global total column CO2 ("XCO2") data set (based on satellite data) covering 14 years
• We present a new method to compute annual mean XCO2 growth rates from this data set
• We present a new annual mean CO2 growth rate time series (covering the entire atmosphere, not only near-surface CO2) including a comparison with growth rates from NOAA based on surface CO2 observations; we find agreement within the reported uncertainty ranges and therefore consider our growth rates to be validated
• We present an answer to the question "Assuming that the variability of the CO2 growth rate is dominated by ENSO and by human emissions, which of the two considered causes dominates the growth rate variability given the satellite-derived growth rates and their uncertainty?" To answer this question we used a statistical analysis method, which we clearly explain. Our answer is given in the Conclusions section: "Our analysis also shows that the ENSO impact on CO2 growth rate variations dominates over that of human emissions throughout the period 2003-2016 (14 years) but in particular during the period 2010-2016 (second half of the investigated time period) due to strong La Niña and El Niño events. We estimate the probability that the impact of ENSO on the variability is larger than the impact of human emissions to be 63% for the time period 2003-2016. If the time period is restricted to 2010-2016 this probability increases to 94%."

However, only the fourth bullet contains any new science results, and this is minimal. It is generally accepted that natural variability in the tropics is the main driver of the atmospheric growth rate, and ENSO is the dominant source of tropical variability. As noted in the IPCC AR5, "the causes of the year-to-year variability observed in the annual atmospheric CO2 accumulation … are estimated with a medium to high confidence to be mainly driven by terrestrial processes occurring in tropical latitudes as inferred from atmospheric CO2 inversions and supported by ocean data and models." Of course, there is a need for attribution studies to better understand the processes driving interannual variability, but the simple analysis presented in this manuscript does not represent substantial new knowledge. It was suggested by Referee #2 that the authors consider publishing in Atmospheric Measurement Techniques (AMT), and I would agree with that suggestion. Indeed, the first three bullets describing the new

knowledge in the manuscript suggest that the manuscript would be better suited for AMT. In its current form, I believe that the manuscript would be a good, short AMT paper. If the authors insist on publishing in ACP, they need to significantly expand the scope of the growth rate analysis, and perhaps include a model to help with the attribution analysis.

Author's reply:
To submit the paper to AMT instead of ACP has been carefully considered before submission to ACP and we have already provided the reason why we think ACP is appropriate in response to earlier comments from the other referees. We agree that it would be very interesting to better address the attribution aspect, but we consider this out of the scope of this paper, as this would require very detailed modelling. Too bad that you do not think that this paper is suitable for ACP, i.e., that you recommend rejection and submission to AMT. Nevertheless, many thanks for taking the time to read our manuscript and for providing a review.

Technical comment

Referee C2:
Page 4, line 23: This line mentions the "(ii) the spatial variability of the XCO2 within the selected region." What region? Is this referring to the analysis of the different latitude bands that was removed?

Author's reply:
Results for latitude bands are now shown in Annex A. Therefore, we kept this. However, for the revised version of the paper we now added this explanation directly after "… within the selected region": "(
[revised manuscript text omitted]

| **2003** | 1.66 (0.76) | 1.99 (0.72) | 1.54 (0.74) | 1.77 (0.62) |
| **2004** | 1.59 (0.30) | 1.52 (0.29) | 1.71 (0.29) | 1.30 (0.23) |
| **2005** | 2.16 (0.28) | 2.51 (0.26) | 1.99 (0.28) | 2.17 (0.22) |
| **2006** | 2.21 (0.27) | 2.13 (0.25) | 2.22 (0.27) | 2.33 (0.21) |
| **2007** | 2.26 (0.27) | 2.33 (0.25) | 2.20 (0.26) | 2.34 (0.21) |
| **2008** | 1.67 (0.29) | 1.60 (0.27) | 1.81 (0.28) | 1.41 (0.20) |
| **2009** | 1.77 (0.30) | 1.75 (0.30) | 1.86 (0.28) | 1.70 (0.21) |
| **2010** | 2.22 (0.29) | 2.67 (0.29) | 2.08 (0.27) | 2.14 (0.20) |
| **2011** | 1.86 (0.28) | 1.69 (0.27) | 1.86 (0.27) | 2.19 (0.19) |
| **2012** | 2.46 (0.29) | 2.64 (0.28) | 2.44 (0.27) | 2.38 (0.21) |
| **2013** | 2.27 (0.30) | 2.38 (0.28) | 2.27 (0.28) | 2.10 (0.22) |
| **2014** | 1.74 (0.31) | 1.53 (0.30) | 1.80 (0.29) | 1.84 (0.23) |
| **2015** | 2.89 (0.34) | 2.89 (0.31) | 2.97 (0.32) | 2.54 (0.25) |
| **2016** | 3.23 (0.50) | 3.28 (0.46) | 3.23 (0.48) | 3.41 (0.36) |